# Efficient Robust Conformal Prediction via Lipschitz-Bounded Networks

**Thomas Massena** [* 1 2]  **Léo Andéol** [* 3 2]  **Thibaut Boissin** [4]  **Franck Mamalet** [4]  **Corentin Friedrich** [4]
**Mathieu Serrurier** [1]  **Sébastien Gerchinovitz** [4 3]

## Abstract

Conformal Prediction (CP) has proven to be an effective post-hoc method for improving the trustworthiness of neural networks by providing prediction sets with finite-sample guarantees. However, under adversarial attacks, classical conformal guarantees do not hold anymore: this problem is addressed in the field of Robust Conformal Prediction. Several methods have been proposed to provide robust CP sets with guarantees under adversarial perturbations, but, for large scale problems, these sets are either too large or the methods are too computationally demanding to be deployed in real life scenarios. In this work, we propose a new method that leverages Lipschitz-bounded networks to precisely and efficiently estimate robust CP sets. When combined with a 1-Lipschitz robust network, we demonstrate that our *lip-rcp* method outperforms state-of-the-art results in both the size of the robust CP sets and computational efficiency in medium and large-scale scenarios such as *ImageNet*. Taking a different angle, we also study vanilla[1]CP under attack, and derive new worst-case coverage bounds of vanilla CP sets, which are valid simultaneously for all adversarial attack levels. Our *lip-rcp* method makes this second approach as efficient as vanilla CP while also allowing robustness guarantees.

## 1. Introduction

With the development of neural networks, and their applications in industrial settings, the study of their reliability has become prevalent. Many approaches have been studied to improve the trustworthiness of neural networks (Delseny et al., 2021). One of them is Uncertainty Quantification (UQ), which has become a key research domain for deploying safety-critical deep learning models. Its purpose is to provide additional information on the output of a model, indicating the confidence in its prediction.

We focus here on a UQ framework called Conformal Prediction (CP), which provides guarantees in nominal settings. It is a finite-sample, distribution-free and model-agnostic framework that efficiently constructs prediction sets which contain the ground truth with high probability. The main underlying assumption is mild: calibration and test data points are assumed to be exchangeable (a lighter assumption than independence and identical distribution) (Vovk et al., 2005). CP has successfully been applied to numerous fields: classification (Sadinle et al., 2019; Ding et al., 2024), regression (Papadopoulos et al., 2011; Romano et al., 2019), object detection (Andéol et al., 2023; Timans et al., 2025), semantic segmentation (Brunekreef et al., 2024; Mossina et al., 2024) and generative models (Mohri & Hashimoto, 2024; Teneggi et al., 2023). Overall, CP represents a powerful tool in order to trust deep learning systems in their nominal settings.

However, research on adversarial robustness has demonstrated that state-of-the-art models can be misled with minimal adversarial input perturbations (Szegedy et al., 2014). This vulnerability has led to extensive research into adversarial robustness, focusing on both attack strategies and defense mechanisms (Goodfellow et al., 2015; Carlini & Wagner, 2017). However, some popular adversarial defense strategies have been shown not to hold in the face of more sophisticated attacks (Athalye et al., 2018). Therefore, research on certifiable robustness aims to provide theoretical worst-case guarantees for model performance under adversarial attacks independently from the attack method. These certified robustness methods also exhibit interesting properties in the domains of Reinforcement Learning (Russo & Proutiere, 2019; Corsi et al., 2020), Differential Privacy (Béthune et al., 2024; Wu et al., 2024), explainable AI (Serrurier et al., 2024; Fel et al., 2023).

Recently, some works in robust Conformal Prediction (Li et al., 2024; Ledda et al., 2023; Liu et al., 2024) have demonstrated that minimal adversarial perturbations can undermine its associated guarantees. Therefore, developing provably robust CP methods is a crucial objective in order to reconcile the guarantees of CP in nominal settings and the worst-case guarantees of certifiably robust systems. Indeed,

---

[*]Equal contribution  [1]IRIT  [2]SNCF  [3]Institut de Mathématiques de Toulouse  [4]IRT Saint Exupery. Correspondence to: Thomas Massena <thomas.massena@irit.fr>.

*Proceedings of the 42nd International Conference on Machine Learning*, Vancouver, Canada. PMLR 267, 2025. Copyright 2025 by the author(s).

---

[1]Throughout this paper we refer to standard CP as vanilla CP.

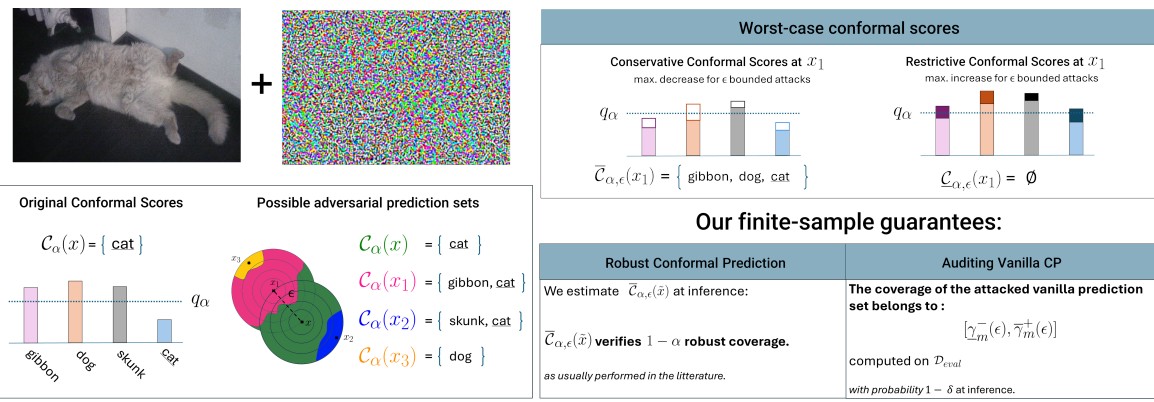

Figure 1: Our framework allows different certifiable guarantees for conformal prediction sets. Through the fast estimation of worst-case conformal prediction scores, the user can either return $\overline{\mathcal{C}}_{\alpha,\epsilon}(\tilde{x})$ which provably verifies Def. 2.2 (§ 4). Or, compute worst-case conformal coverage bounds $\underline{\gamma}_m^-(\epsilon)$ and $\overline{\gamma}_m^+(\epsilon)$ of vanilla CP sets on $\mathcal{D}_{\text{test}}$ through an efficient post-hoc auditing process on a holdout dataset $\mathcal{D}_{\text{eval}}$. This guarantee holds with probability $1 - \delta$ (§ 3). Importantly, the latter's coverage bounds hold over all $\epsilon$ levels.

the vulnerability of CP methods to adversarial attacks raises significant concerns for deploying CP in safety-critical settings, where reliability is essential.

For this purpose, several approaches were proposed to construct robust conformal sets by using randomized smoothing methods (Gendler et al., 2022; Yan et al., 2024) or formal verification solvers (Jeary et al., 2024). These methods inflate the CP sets in order for them to remain valid under adversarial conditions. Therefore, the performance of robust CP methods is measured by their ability to provide small and meaningful CP sets which maintain $1 - \alpha$ conformal coverage under $\epsilon$-bounded perturbations. Unfortunately, these works suffer from two penalties: they lack scalability, and/or provide conservative prediction sets that are too large for practical usage. We address these two shortcomings through the following set of contributions:

1. We complement theoretical guarantees on robust CP from a different angle. We introduce a new method to audit vanilla CP's robustness to adversarial attacks. We derive the first sound coverage bounds for vanilla CP that are valid simultaneously across all attack levels.

2. We provide a simple and efficient method, called *lip-rcp*, to compute lower and upper bounds on CP scores under adversarial attacks. The method applies to any Lipschitz-bounded network, and can be used both to construct robust CP sets and to audit vanilla CP's robustness. It is tailored for efficiency and compatibility with real-time embedded systems.

3. We propose to combine *lip-rcp* with robust 1-Lipschitz networks with tight certified Lipschitz bounds. This allows to obtain tight estimates on worst-case variations of conformal prediction scores.

4. We validate the whole approach across the *CIFAR-10*,

*CIFAR-100*, *Tiny ImageNet* and *ImageNet* datasets. Our experiments showcase negligible computational overhead compared to vanilla CP, with best-in-class performances for both robust CP and vanilla CP's auditing.

## 2. Background and Related Works

### 2.1. Split Conformal Prediction

Multiple approaches to Conformal Prediction exist (Vovk et al., 2005; Angelopoulos & Bates, 2023). We focus on Split Conformal Prediction (Papadopoulos et al., 2002), a variant applied post-hoc using a separate data split. This approach has the advantage of being applicable to any model, even pre-trained on a different dataset. Throughout this work, we focus on classification tasks. We denote $\mathcal{X}$ as the input domain for a classifier $f : \mathcal{X} \to \mathbb{R}^c$ which maps inputs to class logits of labels $y \in \mathcal{Y}$ with $\text{Card}(\mathcal{Y}) = c$ the number of classes. Generally lowercase letters $x, y$ refer to deterministic examples, while uppercase letters $X, Y$ are for random variables. Split CP methods construct prediction sets $\mathcal{C}_\alpha(x)$ that contain the true label with probability at least $1 - \alpha$, where $\alpha \in \left[\frac{1}{n_{\text{cal}}+1}, 1\right)$ is a user-specified risk.[2]

For this purpose, CP defines $s : \mathcal{X} \times \mathcal{Y} \to \mathbb{R}$, a non-conformity score that measures the degree of incorrectness of a prediction $f(x)$ for a ground truth label $y$. On a hold-out "calibration" data split $\mathcal{D}_{\text{cal}} \in (\mathcal{X} \times \mathcal{Y})^{n_{\text{cal}}}$, the non-conformity score is computed for each instance, denoted as $R_i = s(X_i, Y_i)$ for $i \in \{1, \ldots, n_{\text{cal}}\}$, along with the quantile of the score $q_\alpha = \overrightarrow{R}_{\lceil (n_{\text{cal}}+1)(1-\alpha) \rceil}$ (where $\overrightarrow{R}$ represents the $R_i$ scores sorted in ascending order). For a

---

[2]Though some papers might mention the interval $(0, 1)$, this is a slight abuse of notation, since split CP methods are only well defined for $\alpha \geq 1/(n_{\text{cal}} + 1)$.

given example $x_{\text{test}}$, the prediction set is then defined as $\mathcal{C}_\alpha(x_{\text{test}}) = \{y \in \mathcal{Y} : s(x_{\text{test}}, y) \leq q_\alpha\}$. This formulation is quite intuitive: the prediction set $\mathcal{C}_\alpha(x_{\text{test}})$ is defined as all the labels $y$ that would lead to a non-conformity score below $q_\alpha$, and would place among the $1 - \alpha$ most conforming pairs compared to the calibration set. This set then satisfies the following *coverage guarantee*.

**Theorem 2.1** (Vovk et al. 2005)**.** *Let* $\mathcal{D}_{\text{cal}} = \{(X_i, Y_i)\}_{1 \leq i \leq n_{\text{cal}}}$ *and* $(X_{\text{test}}, Y_{\text{test}})$ *be exchangeable random variables. For any non-conformity score* $s : \mathcal{X} \times \mathcal{Y} \to \mathbb{R}$ *and any user-specified risk* $\alpha \in \left[ \frac{1}{n_{\text{cal}}+1}, 1 \right)$, *the prediction set* $\mathcal{C}_\alpha(X_{\text{test}}) = \{y \in \mathcal{Y} : s(X_{\text{test}}, y) \leq q_\alpha\}$ *satisfies:*

$$\mathbb{P}\left(Y_{\text{test}} \in \mathcal{C}_\alpha(X_{\text{test}})\right) \geq 1 - \alpha \,, \tag{1}$$

*where the probability is taken over the random draws of both* $\mathcal{D}_{\text{cal}}$ *and* $(X_{\text{test}}, Y_{\text{test}})$.

This result allows turning predictions from any black-box model $f$ into prediction sets with a guaranteed risk level. However, this guarantee is marginal: the error rate of $\alpha$ only bounds an average error over all possible values of $X_{\text{test}}$.

## 2.2. Robust Conformal Prediction

Extending the CP guarantee of Theorem 2.1 to cover adversarial conditions was first explored in the seminal work of Gendler et al. (2022). In this paper, the following definition of Robust Conformal Prediction is given. For any $x \in \mathcal{X}$, we write $\mathcal{B}_\epsilon(x) = \{x' \in \mathcal{X} : \|x' - x\| \leq \epsilon\}$.

**Definition 2.2** (Robust Conformal Prediction)**.** Let $\mathcal{D}_{\text{cal}} = \{(X_i, Y_i)\}_{1 \leq i \leq n_{\text{cal}}}$ and $(X_{\text{test}}, Y_{\text{test}})$ be exchangeable random variables. A prediction set $\overline{\mathcal{C}}_{\alpha,\epsilon}$ is said to be robust to $\epsilon$-bounded perturbations if for any random variable $\tilde{X}_{\text{test}}$ such that $\tilde{X}_{\text{test}} \in \mathcal{B}_\epsilon(X_{\text{test}})$ almost surely,

$$\mathbb{P}\left[Y_{\text{test}} \in \overline{\mathcal{C}}_{\alpha,\epsilon}(\tilde{X}_{\text{test}})\right] \geq 1 - \alpha. \tag{2}$$

Enforcing Def. 2.2 ensures the coverage guarantee of Theorem 2.1 in worst-case adversarial conditions for any sample under $\epsilon$-bounded adversarial perturbations.

The initial intuition behind robust CP methods is the following: by computing provable lower bounds for conformal prediction scores under attack, it is possible to guarantee robust CP coverage as defined in Def. 2.2. We recall the following concepts, which appear under several wordings in the literature.

**Definition 2.3.** A *conservative score* under $\epsilon$-bounded adversarial perturbations is any function $\underline{s} : \mathcal{X} \times \mathcal{Y} \to \mathbb{R}$ such that, for all $(x, y) \in \mathcal{X} \times \mathcal{Y}$,

$$\underline{s}(x, y) \leq \inf_{\tilde{x} \in \mathcal{B}_\epsilon(x)} s(\tilde{x}, y) \,. \tag{3}$$

**Definition 2.4.** Let $x \in \mathcal{X}$. A *robust prediction set* (or *conservative prediction set*) under $\epsilon$-bounded adversarial perturbations around $x$ is defined as:

$$\overline{\mathcal{C}}_{\alpha,\epsilon}(x) = \{y \in \mathcal{Y} : \underline{s}(x, y) \leq q_\alpha\} \,, \tag{4}$$

where $\underline{s}$ is a conservative score under $\epsilon$-bounded adversarial perturbations.

**General formulation** Importantly, the robust prediction set of Def. 2.4 verifies robust CP coverage as defined in Def. 2.2, and as demonstrated in Jeary et al. (2024). Moreover, as a general rule, robust CP methods revolve around the tight estimation of the conservative score of Eq. (3) in order to verify robust CP coverage. We now discuss the specifics of these computations.

## 2.3. Related Works

**Monte-Carlo methods** In order to compute robust CP sets, the initial work of Gendler et al. (2022) leverages a *smoothed score function* from the framework of randomized smoothing (Cohen et al., 2019). The *RSCP* score is defined as:

$$\tilde{s}(x, y) = \Phi^{-1}\left(\mathbb{E}_{\Delta \in \mathcal{N}(0, \sigma^2 . I)}[s(x + \Delta, y)]\right), \tag{5}$$

with $\Phi^{-1}$ the inverse CDF of the standard normal distribution of standard deviation $\sigma$, which satisfies for any $(x, y) \in \mathcal{X} \times \mathcal{Y}$:

$$\underline{s}_{RSCP}(x, y) = \tilde{s}(x, y) - \frac{\epsilon}{\sigma}, \tag{6}$$

therefore providing a conservative score which will be used to enforce robust CP. However, due to the unknown nature of the expectation in Eq. (5), it is estimated by a Monte Carlo (MC) procedure for which no correction is applied.

Later work (Yan et al., 2024) enables provable robustness coined as the *RSCP+* method by introducing both a corrective factor on the expectation estimation and accounting for the error probability of the MC estimation process by adjusting the calibration threshold. Additionally, this paper introduces *PTT*, an extension to their method with the use of an extra $\mathcal{D}_{\text{holdout}}$ data split to obtain better robust CP metrics. Similarly, the *CAS* method relies on CDF-Aware Sets (Zargarbashi et al., 2024) for certifiably robust CP. These *CAS* sets provide a generally tighter bound for smoothed CP scores than *RSCP*. In an independent work to ours, Zargarbashi & Bojchevski (2025) propose a binary-certificate-based method (*BinCP*) for robust CP with improved sample efficiency and tighter binary certificate bounds.

**Deterministic methods** The authors of *VRCP* (Jeary et al., 2024) leverage formal verification-based solvers (Katz et al., 2017) to provide a deterministic lower bound of Eq. (3) valid locally around $x$. Interestingly, they devise two methods

to achieve robust CP: *VRCP-I* uses a standard vanilla CP calibration procedure, then computes conservative scores around each test point $x_{\text{test}}$. These scores are then used to form the robust CP set $\overline{\mathcal{C}}_{\alpha,\epsilon}(x_{\text{test}})$. *VRCP-C* however computes conservative scores around each pair $(x_i, y_i) \in \mathcal{D}_{\text{cal}}$ and then calibrates the model using these conservative scores to provide a robust CP guarantee which holds at inference and requires no extra computations at test time. Unfortunately, computing *exact* bounds for Eq. (3) is often untractable for neural networks. Therefore, formal verification methods use *incomplete* solvers to compute loose bounds on small to medium size model architectures.

Independently, Ghosh et al. (2023) introduced a novel "quantile of quantiles" method for robust CP based on calibration-time operations. In order to mitigate the set size inflation phenomenon that is inherent to robust CP, the *PRCP* method provides a relaxed "average" coverage guarantee over a predefined noise distribution. These guarantees are applicable on top of most robust CP methods and can be seen as orthogonal to all previously mentioned works, as argued in Zargarbashi et al. (2024). Moreover, complementing previous methods, Aolaritei et al. (2025) propose an extended framework that also provides robustness against global perturbations.

**Scaling robust CP**  Current methods for estimating conservative scores exhibit severe scalability issues. Indeed, verification methods have computational complexities that grow quadratically with the number of neurons and layers of the model. Moreover, randomized smoothing methods require $n_{\text{mc}}$ times more memory at inference time to run an MC estimation process. Also, some theoretical works indicate that the certifiable radii of smoothing methods suffers greatly from high input dimensionality (Wu et al., 2024).

Overall, these different limitations represent an obstacle to scaling robust CP to larger inputs, models and generally harder tasks such as the *ImageNet* dataset for example. In this paper, we present a framework for robust CP with deterministic Lipschitz bounds that has *no such limitations* and avoids time or memory complexity issues at scale.

**Paper Outline.**  The paper is organized as follows. In Section 3 we start by complementing theoretical guarantees on CP's robustness by designing a new auditing process for vanilla CP under attack. We prove coverage bounds that are valid simultaneously across all attack levels. In Section 4, we introduce our *lip-rcp* method and leverage Lipschitz-bounded neural networks to efficiently and accurately approximate local variations of CP scores. This enables efficient robust CP set construction and vanilla CP auditing. Finally, when combined with Lipschitz-by-design networks, we demonstrate the efficiency and superior performances of our approach on several image classification datasets (Section 5).

## 3. Certifiable Coverage Bounds for Vanilla CP Under Attack

In this section, we address the robustness of CP methods to adversarial attacks from a different angle than in Sections 2.2 and 2.3. Instead of enlarging vanilla CP's prediction sets to make them robust by design, we provide tools to reliably evaluate by how much the nominal $1 - \alpha$ coverage of vanilla CP for clean data is impacted under attack. This can be useful when prediction sets of small size are desired, yet knowledge of certifiable coverage guarantees under bounded perturbations are still required. The question of auditing vanilla CP under attacks was already raised earlier (Gendler et al., 2022; Zargarbashi et al., 2024). We provide a rigorous answer below. As a benefit, our guarantee holds simultaneously for all perturbation budgets $\epsilon > 0$.

We show in Section 4 how to *efficiently* implement methods from both approaches (Section 2.2 and this section) when working with Lipschitz-bounded neural networks.

### 3.1. Setting: Vanilla CP Under Attack

Let $h$ be any function that constructs an adversarial example $h(x, \epsilon) \in \mathcal{B}_\epsilon(x)$ from any input $x \in \mathcal{X}$, with an attack budget at most of $\epsilon$, i.e., $\|h(x, \epsilon) - x\| \leq \epsilon$. We define the *coverage under attack* by

$$\gamma_h(\epsilon) = \mathbb{P}_{\mathcal{D}_{\text{test}}}(Y_{\text{test}} \in \mathcal{C}_\alpha(h(X_{\text{test}}, \epsilon)) , \qquad (7)$$

where the probability is taken with respect to the random pair $(X_{\text{test}}, Y_{\text{test}})$. In more mathematical terms, the probability is conditional to $\mathcal{D}_{\text{cal}}$; all statements in this section are valid for any realization of $\mathcal{D}_{\text{cal}}$. Note that here, contrary to robust CP, we consider the vanilla CP set $C_\alpha$.

In the sequel, we show how to provide bounds on $\gamma_h(\epsilon)$. Since the attack $h$ is unknown, we introduce two functions $\underline{\gamma}$ and $\overline{\gamma}$ that can be reliably approximated, and for which $\underline{\gamma}(\epsilon) \leq \gamma_h(\epsilon) \leq \overline{\gamma}(\epsilon)$. To that end, we define both conservative and restrictive prediction sets as follows; $\overline{\mathcal{C}}_{\alpha,\epsilon}(x)$ is a tight version of (4), while $\underline{\mathcal{C}}_{\alpha,\epsilon}(x)$ is new.

**Definition 3.1** (Conservative/Restrictive Prediction Set). Consider the two scores

$$\underline{s}(x, y) = \inf_{\tilde{x} \in \mathcal{B}_\epsilon(x)} s(\tilde{x}, y) \qquad (8)$$

$$\bar{s}(x, y) = \sup_{\tilde{x} \in \mathcal{B}_\epsilon(x)} s(\tilde{x}, y) . \qquad (9)$$

For any $x \in \mathcal{X}$, we define the *conservative prediction set* by $\overline{\mathcal{C}}_{\alpha,\epsilon}(x) = \{y \in \mathcal{Y} : \underline{s}(x, y) \leq q_\alpha\}$, and the *restrictive prediction set* by $\underline{\mathcal{C}}_{\alpha,\epsilon}(x) = \{y \in \mathcal{Y} : \bar{s}(x, y) \leq q_\alpha\}$.

When $x \mapsto s(x, y)$ is Lipschitz, both sets can be efficiently (and provably) approximated, as shown in Section 4.

We are now ready to introduce the two functions $\underline{\gamma}$ and $\overline{\gamma}$,

defined for all $\epsilon \geq 0$ by

$$\underline{\gamma}(\epsilon) = \mathbb{P}_{\mathcal{D}_{\text{test}}}\big(Y_{\text{test}} \in \underline{\mathcal{C}}_{\alpha,\epsilon}(X_{\text{test}})\big) \tag{10}$$

$$\overline{\gamma}(\epsilon) = \mathbb{P}_{\mathcal{D}_{\text{test}}}\big(Y_{\text{test}} \in \overline{\mathcal{C}}_{\alpha,\epsilon}(X_{\text{test}})\big) \tag{11}$$

Noting that $\underline{s}(x,y) \leq s(h(x,\epsilon),y) \leq \bar{s}(x,y)$ and therefore $\underline{\mathcal{C}}_{\alpha,\epsilon}(\cdot) \subseteq C_\alpha(h(\cdot,\epsilon)) \subseteq \overline{\mathcal{C}}_{\alpha,\epsilon}(\cdot)$, we can see that $\underline{\gamma}(\epsilon) \leq \gamma_h(\epsilon) \leq \overline{\gamma}(\epsilon)$ for all $\epsilon \geq 0$. Therefore, though not used by vanilla CP, the sets $\underline{\mathcal{C}}_{\alpha,\epsilon}(X_{\text{test}})$ and $\overline{\mathcal{C}}_{\alpha,\epsilon}(X_{\text{test}})$ are instrumental in controlling the coverage under attack.

## 3.2. A Provable and Tight Control on $\underline{\gamma}$ and $\overline{\gamma}$

The question of bounding $\underline{\gamma}(\epsilon)$ from below, i.e., guaranteeing a minimal coverage of vanilla CP under attack, already appeared in Theorem 2 by Gendler et al. (2022). We however realized that a subtle mathematical mistake (of an overfitting flavor) was unfortunately left aside, which impacts the correctness of their guarantee; see Appendix A for details. The same mistake was reproduced in the second part of Proposition 5.1 by Zargarbashi et al. (2024).

To overcome this overfitting-type issue, we work with an additional dataset $\mathcal{D}_{\text{eval}}$ of $m$ data points drawn i.i.d. from the same distribution but independently from $\mathcal{D}_{\text{cal}}$. Denote these points by $(X_i, Y_i)_{1 \leq i \leq m}$.

Let $\mathbb{1}$ denote the indicator function. We define empirical counterparts (based on $\mathcal{D}_{\text{eval}}$) of $\underline{\gamma}(\epsilon)$ and $\overline{\gamma}(\epsilon)$ as follows: for all $\epsilon \geq 0$,

$$\overline{\gamma}_m(\epsilon) = \frac{1}{m}\sum_{i=1}^{m} \mathbb{1}\{Y_i \in \overline{\mathcal{C}}_{\alpha,\epsilon}(X_i)\} \tag{12}$$

$$\underline{\gamma}_m(\epsilon) = \frac{1}{m}\sum_{i=1}^{m} \mathbb{1}\{Y_i \in \underline{\mathcal{C}}_{\alpha,\epsilon}(X_i)\} \tag{13}$$

To account for statistical deviations, next we work with slightly corrected estimators studied earlier, e.g., by Langford & Schapire (2005). For $m \geq 1$, $p \in [0,1]$, and $0 \leq k \leq m$, we write $F_{m,p}(k) = \sum_{j=0}^{k}\binom{m}{j}p^j(1-p)^j$ for the cumulative distribution function of the Binomial$(m,p)$ distribution. For any attack budget $\epsilon \geq 0$ and any risk level $\delta \in (0,1)$, we define

$$\overline{\gamma}_m^+(\epsilon,\delta) = \max\Big\{p \in [0,1] : F_{m,p}\big(m\overline{\gamma}_m(\epsilon)\big) \geq \delta\Big\} \tag{14}$$

$$\underline{\gamma}_m^-(\epsilon,\delta) = 1-\max\Big\{p \in [0,1]: F_{m,p}\big(m(1-\underline{\gamma}_m(\epsilon))\big) \geq \delta\Big\} \tag{15}$$

Though technical at first sight, these estimators are tailored for the binomial distribution and thus tighter than, e.g., high-probability bounds obtained from Hoeffding's bound. Given $\overline{\gamma}_m(\epsilon)$ and $\underline{\gamma}_m(\epsilon)$, they can be efficiently computed with a binary search (by monotonicity of $p \mapsto F_{m,p}(k)$).

Next we work under the following mild assumptions on the input space $\mathcal{X}$ and the non-conformity score $s(x,y)$.

**Assumption 3.2.**

(A1) $\mathcal{X} \subset \mathbb{R}^d$ is convex and closed (for some $d \geq 1$);

(A2) $x \in \mathcal{X} \mapsto s(x,y)$ is continuous for all $y \in \mathcal{Y}$.

The first condition (A1) is satisfied, e.g., for classical vector spaces used to represent images (such as $\mathbb{R}^{H\cdot W\cdot 3}$, with $H$ and $W$ respectively the height and width of the images). Importantly, the assumption is on the underlying space $\mathcal{X}$ and not on the probability distribution over it. Therefore, in our application setting, it holds true even if the distribution of images has a nonconvex support. The second condition (A2) is satisfied whenever the score is of the form $s(x,y) = \psi(f(x),y)$, for a continuous model $f$ and some continuous function $u \mapsto \psi(u,y)$. In our setting, the assumption always holds true, due to both $f$ and $s(\cdot,y)$ being Lipschitz-continuous.[3]

The main result of this section is the following.

**Theorem 3.3.** *Suppose that Assumption 3.2 holds true. Assume also that $\mathcal{D}_{\text{cal}}, \mathcal{D}_{\text{eval}}, \mathcal{D}_{\text{test}}$ are made of i.i.d. pairs $(X_i, Y_i)$. Let $q_\alpha$ be the empirical quantile computed by vanilla CP using $\mathcal{D}_{\text{cal}}$, and let $m \geq 2$ be the number of points in $\mathcal{D}_{\text{eval}}$.*

*Let $\delta \in (0,1)$ and set $\delta' = \delta/(2m-2)$. Then, for any attack function $h$ and almost every $\mathcal{D}_{\text{cal}}$,*

$$\mathbb{P}_{\mathcal{D}_{\text{eval}}}\Big(\forall \epsilon > 0, \ \underline{\gamma}_m^-(\epsilon,\delta') \leq \gamma_h(\epsilon) \leq \overline{\gamma}_m^+(\epsilon,\delta')\Big) \geq 1 - \delta, \tag{16}$$

*where the probability is over the random draw of $\mathcal{D}_{\text{eval}}$.[4]*

The proof follows by combining $\underline{\gamma}(\epsilon) \leq \gamma_h(\epsilon) \leq \overline{\gamma}(\epsilon)$ with Theorems B.2 and B.3 in Appendix B (applied with the risk level $\delta/2$), followed by a union bound. These theorems are similar in spirit to the DKW inequality (Dvoretzky et al., 1956; Massart, 1990) or variants (Vapnik & Chervonenkis, 1974; Anthony & Shawe-Taylor, 1993), which are useful concentration inequalities to estimate an unknown cumulative distribution function based on i.i.d. samples. In particular, we borrow arguments from Ducoffe et al. (2020, Theorem 1), combined with a careful treatment of discontinuity points of $\underline{\gamma}$ and $\overline{\gamma}$.

**Interpretation** Our result provides a probabilistic guarantee for robustness under adversarial attacks of arbitrary budgets. Specifically, for *any* calibration set $\mathcal{D}_{\text{cal}}$ and *any* attack function $h$ (regardless of the norm), the coverage under attack $\gamma_h$ is bounded by $\underline{\gamma}_m^-$ and $\overline{\gamma}_m^+$ with probability $1 - \delta$ over the randomness in the holdout set $\mathcal{D}_{\text{eval}}$, simultaneously for all perturbation budgets $\epsilon > 0$. This

---

[3]Note that, in this section, we do not assume that $x \mapsto s(x,y)$ is smooth (beyond continuity), contrary to scores obtained after randomized smoothing.

[4]In more formal terms, the probability $\mathbb{P}_{\mathcal{D}_{\text{eval}}}(\cdot)$ is conditional to $\mathcal{D}_{\text{cal}}$, and our inequality holds almost surely.

simultaneity strengthens the practical utility of the bounds, as no prior assumptions about the adversary's strategy are required. Moreover, this result holds not only for the exact sets $\underline{\mathcal{C}}_{\alpha,\epsilon}$ and $\overline{\mathcal{C}}_{\alpha,\epsilon}$, but also for any $\underline{\mathcal{C}}'_{\alpha,\epsilon}$ and $\overline{\mathcal{C}}'_{\alpha,\epsilon}$ verifying $\underline{\mathcal{C}}'_{\alpha,\epsilon} \subseteq \underline{\mathcal{C}}_{\alpha,\epsilon}$ and $\overline{\mathcal{C}}_{\alpha,\epsilon} \subseteq \overline{\mathcal{C}}'_{\alpha,\epsilon}$. This allows to use approximate sets from common robustness approaches, such as that of Section 4. Naturally, the informativeness of the bound is directly linked to the tightness of the estimate.

# 4. Fast Computations for Robust and Vanilla CP with Lipschitz bounds

In this section, we introduce Lipschitz bounds for neural networks and use their validity across the totality of the support of $\mathcal{X}$ to enable fast computations of conservative and restrictive scores.

## 4.1. On the Lipschitz Constant of Neural Networks

**Definition 4.1** (Lipschitz constant). A neural network classifier $f : \mathcal{X} \to \mathbb{R}^c$ is said to be $L$-Lipschitz in $\ell_p$ norm if it verifies the following behavior:

$$\forall (x,y) \in \mathcal{X}^2, \|f(x) - f(y)\|_p \leq L.\|x - y\|_p \quad (17)$$

Importantly, most deep neural networks -excluding attention based architectures (Havens et al., 2024)- are Lipschitz continuous.

**Estimating Lipschitz constants of neural networks** Computing the exact Lipschitz constant of deep neural networks is an NP-hard problem as stated in (Virmaux & Scaman, 2018). In order to mitigate this issue, some methods like (Wang et al., 2024) compute over-estimations of the Lipschitz constant of the network, for instance by computing the product of the Lipschitz constant of the network's layers. Unfortunately, these methods currently offer very loose estimations, any breakthroughs in that field would benefit our framework.

A more popular alternative to ad-hoc computation lies in the field of "Lipschitz by design" architectures. This very active field relies on the seminal work of (Anil et al., 2019) to provide efficient weight re-parametrizations that ensure 1-Lipschitz behavior in $\ell_2$ norm (in general). Some notable examples include Riemannian optimization inspired implementations (Lezcano Casado, 2019), or even performance-focused implementations with reduced computational over-heads during training (Araujo et al., 2023; Boissin et al., 2025). Using Lipschitz-constrained networks introduces the several advantages. First and foremost, Lipschitz-constrained networks allow *explicit* control of their placement on the robustness-accuracy trade-off (Béthune et al., 2022). Also, additional orthogonality constraints on these networks have been shown to result in generally tighter Lip-

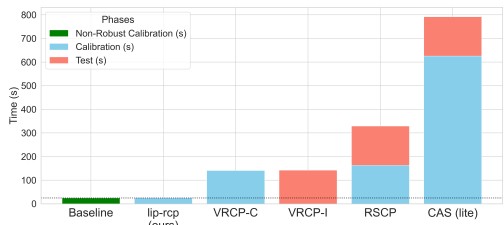

Figure 2: Calibration and test time on the *CIFAR-10* dataset for different robust CP methods. Our method has negligible overhead compared to vanilla CP. Here we use $n_{\text{cal}} = 4750$, $n_{\text{test}} = 4750$ and $n_{\text{mc}} = 1024$, *CAS (lite)* corresponds to the version of *CAS* that leverages CDF-Aware smoothed Prediction Sets only at calibration time.

schitz bounds. Finally, these constraints also mitigate the gradient vanishing problem (Li et al., 2019) and allow for more efficient training.

Therefore, Lipschitz-constrained networks have two main advantages: their training objective explicitly promotes robustness and the estimation of the Lipschitz constant is locally tight. These particular traits will be key to getting fast and tight estimations of the conservative and restrictive scores.

## 4.2. Computing Conservative and Restrictive Conformal Scores

In this section, we use the global Lipschitz bound of our networks to efficiently estimate the conservative and restrictive prediction sets at model inference. We call our method *lip-rcp*. These estimations are then used to construct robust CP sets and audit the robustness of vanilla CP as per § 3. Our approach can be connected to the Lipschitz properties exhibited by the smoothed classifier introduced in *RSCP*.

However, at inference time, most Lipschitz weight re-parametrization schemes can be exported as a set of regular neural network weights to eliminate any computational overhead compared to their unconstrained counterparts.

**Lipschitz score bounds** Assuming the non-conformity score $s(x,y)$ can be expressed as $\psi(f(x), y)$ such that $f$ is an $L_n$-Lipschitz classifier and $\psi(\cdot, y)$ is $L_s$-Lipschitz in $\ell_p$ norm for all $y \in \mathcal{Y}$, we can write:

$$\forall y \in \mathcal{Y}, |s(x,y) - s(x + \delta, y)| \leq L_n \cdot L_s \cdot \|\delta\|_p. \quad (18)$$

This induces the following bounds $\forall \tilde{x} \in \mathcal{B}_\epsilon(x), \forall y \in \mathcal{Y}$ :

$$\underbrace{s(x,y) - L_n \cdot L_s \cdot \epsilon}_{=\underline{s}_{lip\text{-}rcp}(x,y)} \leq s(\tilde{x},y) \leq \underbrace{s(x,y) + L_n \cdot L_s \cdot \epsilon}_{=\overline{s}_{lip\text{-}rcp}(x,y)} \quad (19)$$

**Score function** In the literature of CP, the Least Ambiguous Classifier (LAC) score, usually based on the softmax function, is classically used to provide conformal scores.

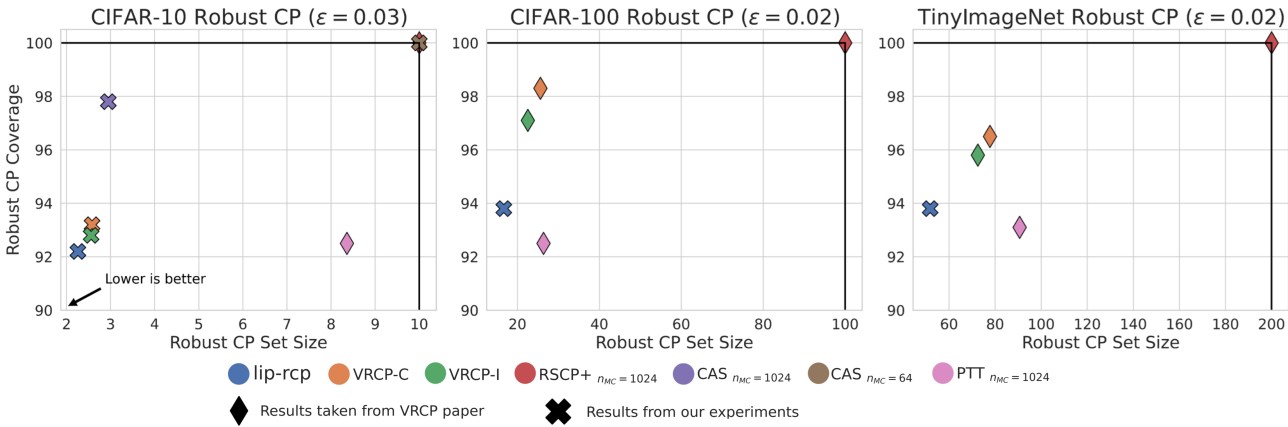

Figure 3: Robust CP set sizes and empirical coverage values across different classification datasets for $\alpha = 0.1$. Ideally, the best robust CP sets are small while nearly achieving the desired coverage guarantee (bottom left corner). We plot values that were taken from the *VRCP* paper with diamonds while our measurements are plotted as crosses.

One alternative we propose is to use a *sigmoid* based conformal prediction score that we call the LAC sigmoid score:

$$s(x, y) = 1 - \text{sigmoid}\left(\frac{f(x)_y - b}{T}\right). \qquad (20)$$

Importantly, the Lipschitz constant of this score function is $L_s = 1/(4 \times T)$ w.r.t $f(x)_y$.

Note that this score differs from the *PTT* score of (Gendler et al., 2022) given that the value of $f(x)_y$ is directly passed through the sigmoid of temperature $T$ and bias $b$. No ranking transformation requiring an extra data split is applied. Furthermore, replacing $s(x, y)$ with $1 - \text{softmax}(f(x)/T)_y$ would be similar in spirit. However, some experiments suggest that it might lead to larger prediction sets; see Appendix E.

**Unified calibration and test-time process** The certificates provided by *VRCP* hold locally around data points $(X_i, Y_i)_{\mathcal{D}_{\text{cal}}}$, and *VRCP* distinguishes two possible robust CP procedures: robust calibration (*VRCP-C*) or robust inference (*VRCP-I*). However, using global Lipschitz bounds eliminates that need. Indeed, a robust calibration threshold $q_{\alpha, \epsilon}$ defined on scores $\underline{s}_{lip\text{-}rcp}(x, y)$ of Eq. (19) yields the same results as robust inference given that the quantile computation is translation equivariant.

**Efficient computation** To estimate the conservative and restrictive scores of Def. 3.1, we can simply leverage the expression of the bounds of Eq. 19. We detail the time complexities for the computation of a non-conformity by different certifiably robust CP methods in Table 1. In addition, we provide runtimes for the calibration and testing steps of different methods in Fig. 2.

| Method | Cal. complexity | Test complexity |
|---|---|---|
| RSCP | $\mathcal{O}(n_{\text{mc}})$ | $\mathcal{O}(n_{\text{mc}})$ |
| CAS | $\mathcal{O}(n_{\text{mc}})$ | $\mathcal{O}(n_{\text{mc}} \times t_b)$ |
| VRCP-I | $\boldsymbol{\mathcal{O}}(1)$ | $\mathcal{O}(t_v)$ |
| VRCP-C | $\mathcal{O}(t_v)$ | $\boldsymbol{\mathcal{O}}(1)$ |
| lip-rcp (ours) | $\boldsymbol{\mathcal{O}}(1)$ | $\boldsymbol{\mathcal{O}}(1)$ |

Table 1: Time complexity for computing a single non-conformity score. Here $n_{\text{mc}}$ is the number of Monte Carlo samples, $t_b$ the CAS-bound cost, and $t_v$ the VRCP solver cost. *lip-rcp* is the only method with constant-time calibration and inference.

## 5. Empirical Validation

To validate the performance of our method, we test our framework across multiple classification datasets. Our networks are composed of a 1-Lipschitz feature extractor followed by a classification layer that ensures that every output respects a 1-Lipschitz condition. We provide information about of the computational overhead of training Lipschitz-constrained networks and the model architectures used for other methods in Appendix F. Finally, our code will be made available on the following github repository.

**Training scheme** We leverage networks with Lipschitz and orthogonality constraints from the library introduced in Boissin et al. (2025) to balance both performance and minimal training overhead. More details are provided in Appendix G.

### 5.1. Robust CP Comparison

We evaluate our method on the *CIFAR-10*, *CIFAR-100* and *TinyImageNet* datasets. Our methodology follows that of the benchmark of *VRCP* and we adopt the same calibration, holdout and test set sizes as Jeary et al. (2024) on

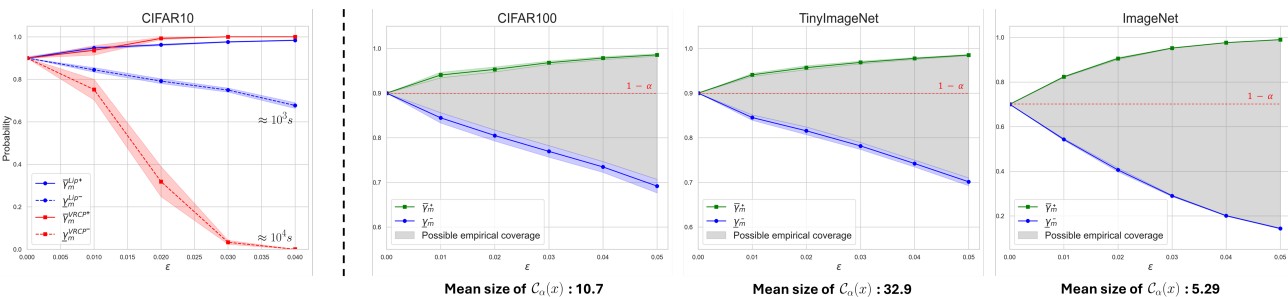

Figure 4: (Left): Comparison between vanilla CP coverage bounds using Lipschitz bounds or the *CROWN* method with $\delta = 10^{-4}$. Approximate computation times are also provided. (Right): Corrected $\overline{\gamma}_m^+$ and $\underline{\gamma}_m^-$ bounds for vanilla CP methods under $\epsilon$ bounded adversarial noise computed on 1-Lipschitz networks given $\delta = 10^{-4}$. Mean prediction set size on clean data is indicated below each plot. We take $n_{cal} = 15000$ and $n_{eval} = 35000$ for ImageNet.

all these datasets. Also, we give the mean values of the robust CP set sizes and the conformal coverage of these robust sets across 25 different random samplings of $\mathcal{D}_{cal}$ and $\mathcal{D}_{test}$ (as well as $\mathcal{D}_{holdout}$ for *PTT*) that were unseen during training. We report our results in Fig. 3 where we also plot the measurements made by Jeary et al. (2024) with different markers. To allow fair comparison with verification methods, we train specific neural networks for *VRCP* on the *CIFAR-10* dataset according to Appendix B of the associated paper. This allows the *CROWN* (Zhang et al., 2018) method to run efficiently on this model. For *CAS*, we use ResNet50 networks. Finally, we do not benchmark the *PRCP* method whose robustness guarantees are specific to the type of adversarial perturbation.

**Interpretation** Our method outperforms existing robust CP approaches across all tested datasets, Fig. 3, it provides smaller robust CP sets while maintaining conformal coverage close to $1 - \alpha$. We give the following explanations to these results. First, the Lipschitz-constrained model training approach we use is more efficient at promoting robustness. Also, neural networks with orthogonality constraints allow tight estimations of their conservative scores. Additionally, smoothing methods suffer from finite sample MC estimations in more than one way: as both a corrective factor is added to the smoothed scores, and the error probability of the MC estimation also necessitates adjusting the quantile computation. Finally, smoothing methods require an unrealistic number of MC samplings to obtain meaningful sets.

**Scaling to the ImageNet dataset** In order to evaluate the scalability of our method, we also validate it on the *ImageNet* dataset. Given the poor scalability of formal verification methods, we were not able to apply *VRCP* on the *ImageNet* dataset. We therefore compare our method to *CAS* on a ResNet50 network, being generally the best performing competing approach. Our model is described in Appendix G.4.

Finally, we used a restricted set of $n = 500$ data points to evaluate the *CAS* method as done in Zargarbashi et al. (2024), given its intensive computational budget. For both methods we use 40% of the data points for calibration and the rest for testing. The obtained results are given in Table 2.

| **Method** | **Set size** | **Cov. (%)** | $n$ | **Time (s)** |
|---|---|---|---|---|
| *CAS* | 1000.0 | 100.0 | $5 \cdot 10^2$ | 7.920 |
| *lip-rcp* | **111.0** | **97.4** | $5 \cdot 10^4$ | 0.012 |

Table 2: Robust CP set sizes and empirical test coverage at $\epsilon = 0.02$, $\alpha = 0.1$ on ImageNet (using *CAS* with $n_{mc} = 1024$). One additional run of our method at $n = 5 \cdot 10^2$ yielded a set size of 118.5 and 97.5% coverage. The time is given *per sample*.

## 5.2. Certifiable Vanilla CP Coverage Bounds Results

In this section, we compute respectively lower and upper bounds of $\underline{\gamma}_m^-$ and $\overline{\gamma}_m^+$ introduced in Section 3 by both Lipschitz and formal verification methods. We first perform vanilla split CP on $\mathcal{D}_{cal}$ consisting of $n_{cal} = 3000$ samples. Next, we compute empirical approximations $\overline{\gamma}_m$ and $\underline{\gamma}_m$ as in (12) and (13) on an evaluation dataset $\mathcal{D}_{eval}$ with $n_{eval} = 5000$ samples with both Lipschitz bounds and the *CROWN* formal verification method. We then compute the corrected estimates $\overline{\gamma}_m^+$ and $\underline{\gamma}_m^-$ as in (14) and (15) guaranteed in Theorem 3.3 by binary search. We repeat this process over 20 randomly sampled, non-overlapping pairs of $\mathcal{D}_{cal}$ and $\mathcal{D}_{eval}$, and plot the mean value with an uncertainty band of 1 standard deviation. The resulting coverage metrics are shown on Fig. 4 (left) on the *CIFAR-10* dataset for both computation methods with their associated models.

We further compute the worst-case conformal coverage values of standard conformal prediction (CP) sets under $\epsilon$-bounded perturbations across all previously mentioned datasets. Due to the larger scale of these tasks, we are only able to compute the bounds for 1-Lipschitz models. The results are presented in Fig. 4 (right).

**Interpretation** As illustrated in Fig. 4 (left), Lipschitz-constrained models yield less pessimistic bounds than verification for the worst-case coverage of vanilla CP under $\epsilon$-bounded adversarial perturbations. This can be explained by the limited scalability of formal verification methods to larger neural networks. Also, Fig 4 (right) uncovers how providing coverage bounds for CP methods applied to robust networks conserves the small and informative nature of CP sets while providing error guarantees under bounded attacks.

## 6. Discussion

**Conclusion** In this work, we propose a novel method for fast computation of certifiably robust prediction sets, improving over SoTA methods both in terms of speed and conformal set sizes. First, through a careful analysis of vanilla CP under attack, we provide novel high-probability bounds on its coverage under attack. Our novel guarantees are valid simultaneously for all attack budgets $\epsilon$ and function $h(\cdot, \epsilon)$, therefore not requiring assumptions on the attacker. Then, we use Lipschitz-bounded networks to estimate robust prediction sets, benefiting from three main advantages: better scalability, better overall robustness and the ability to tightly estimate worst-case variations with little computational overhead. We apply our *lip-rcp* approach not only to compute efficiently bounds for vanilla CP under attack, but also for robust CP. Finally we validate our approaches on multiple classification datasets achieving best-in-class performance with similar computational requirements as vanilla CP.

**Limitations and future work** The method and results presented in this paper rely on some conditions on the model and perturbations, which would be worth generalizing.

Our approach delivers strong certifiable guarantees under $\ell_2$ perturbations but does not generalize to other norms, such as $\ell_1$ or $\ell_\infty$. Integrating advances from works like Biswas (2024) or Zhang et al. (2021) could broaden its applicability and is left for future work. Moreover, while our method is applicable to any network whose Lipschitz constant is computable, we limit our study to Lipschitz-by-design architectures—whose Lipschitz constant is tightly controlled—to enable efficient robust conformal prediction. Note that this approach is not fully model-agnostic as other competing conformal methods, but that tight Lipschitz estimation would restore model agnosticism and pave the way for efficient robust conformal prediction in classification *and regression* settings.

Finally, it would be interesting to investigate whether our approach generalizes to higher values of $\epsilon$ as, e.g., in the works of Gendler et al. (2022); Zargarbashi et al. (2024), while retaining our guarantees and computational efficiency.

## Impact Statement

This paper presents work whose goal is to advance the field of Machine Learning. There are many potential societal consequences of our work, none which we feel must be specifically highlighted here.

## Acknowledgements

The authors would like to thank Agustin Martin Picard for his insights, along with Arthur Chiron and Luca Mossina for their careful proofreading.

This work was carried out within the DEEL project,[5] which is part of IRT Saint Exupéry and the ANITI AI cluster. The authors acknowledge the financial support from DEEL's Industrial and Academic Members and the France 2030 program – Grant agreements n°ANR-10-AIRT-01 and n°ANR-23-IACL-0002.

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

| Method | Certifiable | Fast cal. | Fast test | $\ell_1, \ell_\infty$ | Poisoning | Nb hyperparams |
|---|---|---|---|---|---|---|
| *aPRCP* (Ghosh et al., 2023) | ✗ | ✗ | ✓ | ✗ | ✗ | 2 |
| *RSCP* (Gendler et al., 2022) | ✗ | ✗ | ✗ | ✓ | ✗ | 3 |
| *RSCP+* (PTT/RCT) (Yan et al., 2024) | ✓ | ✗ | ✗ | ✓ | ✗ | 4(6) |
| *CAS* (Zargarbashi et al., 2024) | ✓ | ✗ | ✗ | ✓ | ✓ | 2 |
| *VRCP-I* (Jeary et al., 2024) | ✓ | ✓ | ✗ | ✓ | ✗ | 2 |
| *VRCP-C* (Jeary et al., 2024) | ✓ | ✗ | ✓ | ✓ | ✗ | 2 |
| *lip-rcp* | ✓ | ✓ | ✓ | ✗ | ✓ | 2 |

Table 3: Comparison of certified robustness methods. Note that aPRCP is not certifiable without assumptions on the adversarial distribution. See Section F.2 for details on hyperparameter counts.

# A. On the Lower Bound of Gendler et al. (2022, Theorem 2)

Unfortunately, it seems that a subtle mathematical mistake was left aside in Theorem 2 by Gendler et al. (2022). It can be seen in the statement and in the proof.

In the statement, the deterministic quantity $\mathbb{P}\big[Y_{n+1} \in \mathcal{C}(\tilde{X}_{n+1})\big]$ is bounded from below by the random quantity $\tau$ (note that $\tau$ depends on the calibration data); this lower bound can thus fail in general.[6]

In the proof, which appears in Gendler et al. (2022, Appendix S1, Proof of theorem 2), the authors use Lemma 2 by Romano et al. (2019) which only holds for deterministic values of $\tau$ (or $\alpha$, following the notation in Romano et al. 2019). Since $\tau$ is random, their lemma can unfortunately not be used here.

# B. Proof of Theorem 3.3

Let $\mathcal{X} \subset \mathbb{R}^d$ (Borel subset) and $\|\cdot\|$ a norm on $\mathbb{R}^d$. Let also $\mathcal{Y} = \{1, \cdots, K\}$ and $s : \mathcal{X} \times \mathcal{Y} \to \mathbb{R}$ be a measurable function (non-conformity score).

For any $x \in \mathcal{X}$ and $\epsilon \geq 0$, we set $\mathcal{B}_\epsilon(x) := \{\tilde{x} \in \mathcal{X} : \|\tilde{x} - x\| \leq \epsilon\}$.

There are some minor measurability subtleties in the paper and the following proof, which are overlooked here for readability, but discussed in Section C.

## B.1. On the continuity of $\overline{\gamma}$ and $\underline{\gamma}$

**In this section and the next** We assume that $\mathcal{D}_{\text{cal}}$ is fixed, so that $s(\cdot, \cdot)$ and $q_\alpha$ are deterministic. All probabilities or expectations below are taken w.r.t. the random draws of $\mathcal{D}_{\text{eval}}$ and/or $(X_{\text{test}}, Y_{\text{test}})$. [7] Furthermore, since in our paper $(\mathcal{D}_{\text{cal}})$ is independent from $(\mathcal{D}_{\text{eval}}, (X_{\text{test}}, Y_{\text{test}}))$, all inequalities below translate into inequalities that are valid for a.e. $\mathcal{D}_{\text{cal}}$, provided that $\mathbb{P}(\cdot)$ and $\mathbb{E}[\cdots]$ are replaced with $\mathbb{P}(\cdots|\mathcal{D}_{\text{cal}})$ and $\mathbb{E}[\cdots|\mathcal{D}_{\text{cal}}]$.

We now study $\overline{\gamma}$ and $\underline{\gamma}$. For $\epsilon \geq 0$, recall that

$$\overline{\gamma}(\epsilon) = \mathbb{P}\left(\underline{s}_\epsilon(X_{\text{test}}, Y_{\text{test}}) \leq q_\alpha\right) \quad \text{and} \quad \underline{\gamma}(\epsilon) = \mathbb{P}\left(\bar{s}_\epsilon(X_{\text{test}}, Y_{\text{test}}) \leq q_\alpha\right) ,$$

where, in order to emphasize the dependency in $\epsilon$, we wrote

$$\underline{s}_\epsilon(x, y) = \inf_{\tilde{x} \in \mathcal{B}_\epsilon(x)} s(\tilde{x}, y) \quad \text{and} \quad \bar{s}_\epsilon(x, y) = \sup_{\tilde{x} \in \mathcal{B}_\epsilon(x)} s(\tilde{x}, y) .$$

**Proposition B.1.** *Assume that A1 and A2 from 3.2 hold true. Then, $\overline{\gamma}$ is right-continuous on $[0, +\infty)$ and $\underline{\gamma}$ is left-continuous on $(0, +\infty)$.*

*Proof.* We start with $\overline{\gamma}$. Let $\epsilon \geq 0$ and $\eta > 0$. We show that there exists $\delta > 0$ s.t. $\overline{\gamma}(\epsilon + \delta) \leq \overline{\gamma}(\epsilon) + \eta$, which will (by monotonicity of $\overline{\gamma}$) imply that $|\overline{\gamma}(\epsilon') - \overline{\gamma}(\epsilon)| \leq \eta$ for all $\epsilon' \in [\epsilon, \epsilon + \delta]$.

---

[6]This could make sense if the probability $\mathbb{P}\big[Y_{n+1} \in \mathcal{C}(\tilde{X}_{n+1})\big]$ were conditional on the calibration data, but it is not the case in the proof: the probability is with respect to the joint distribution of $(X_i, Y_i)_{1 \leq i \leq n+1}$.

[7]In fact, we also work conditionally to the training set, which is treated as deterministic in all the paper.

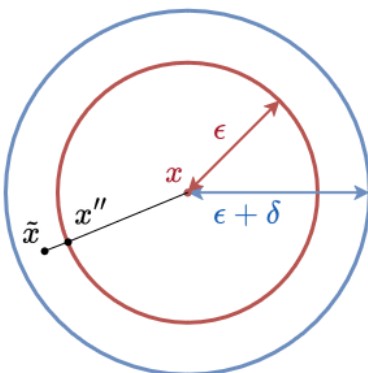

Figure 5: Illustration of the proof of (22). On the ball $\mathcal{B}_{\epsilon+\delta}(x)$, the score $x \mapsto s(x,y)$ is minimized at some $\tilde{x}$. On the smaller ball $\mathcal{B}_\epsilon(x)$, the minimum can only be larger, but not larger than $s(x'',y)$ which is close to $s(\tilde{x},y)$ by continuity of $s(\cdot,y)$.

Let $K \subset \mathcal{X}$ be any compact set such that $\mathbb{P}(X_{\text{test}} \in K) \geq 1 - \frac{\eta}{2}$.

By right-continuity of the cdf $t \mapsto F_\epsilon(t) := \mathbb{P}(\underline{s}_\epsilon(X_{\text{test}}, Y_{\text{test}}) \leq t | X_{\text{test}} \in K)$, there exists $\rho > 0$ s.t. $F_\epsilon(q_\alpha + \rho) \leq F_\epsilon(q_\alpha) + \frac{\eta}{2}$.

We now use the fact that, for each $y \in \mathcal{Y}$ (with $\mathcal{Y}$ finite), the function $s(\cdot, y)$ is continuous and thus uniformly continuous on the compact set $K_{\epsilon+1} := \{x + u : x \in K, u \in \mathbb{R}^d, \|u\| \leq \epsilon + 1\}$. Therefore, there exists $\delta \in (0,1)$ s.t.,

$$\forall y \in \mathcal{Y}, \forall x, x' \in K_{\epsilon+1}, \|x - x'\| \leq \delta \implies |s(x,y) - s(x',y)| \leq \rho. \tag{21}$$

Note that, for any $y \in \mathcal{Y}$ and $x \in K$, by (21) above,

$$\underline{s}_\epsilon(x,y) = \inf_{\tilde{x} \in \mathcal{B}_\epsilon(x)} s(\tilde{x},y) \leq \inf_{\tilde{x} \in \mathcal{B}_{\epsilon+\delta}(x)} s(\tilde{x},y) + \rho = \underline{s}_{\epsilon+\delta}(x,y) + \rho. \tag{22}$$

This follows from Figure 5. More formally, let $\tilde{x} \in \mathcal{B}_{\epsilon+\delta}(x)$ be such that $s(\tilde{x},y) = \underline{s}_{\epsilon+\delta}(x,y)$. Inequality (22) is immediate if $\|\tilde{x} - x\| \leq \epsilon$. We thus assume without loss of generality that $\|\tilde{x} - x\| > \epsilon$. Then, let $x'' \in [x, \tilde{x}] \subset \mathcal{X}$ be such that $\|x'' - x\| = \epsilon$ (recall that $\mathcal{X}$ is convex). In this case $\|\tilde{x} - x''\| \leq \delta$ and thus (by (21)) $s(x'',y) \leq s(\tilde{x},y) + \rho$ which implies $\underline{s}_\epsilon(x,y) \leq s(x'',y) \leq \underline{s}_{\epsilon+\delta}(x,y) + \rho$. This concludes the proof of (22).

We are ready to conclude:

$$\begin{aligned}
\overline{\gamma}(\epsilon + \delta) &= \mathbb{P}\left(\underline{s}_{\epsilon+\delta}(X_{\text{test}}, Y_{\text{test}}) \leq q_\alpha\right) \\
&\leq \mathbb{P}(X_{\text{test}} \in K) \cdot \mathbb{P}\left(\underline{s}_{\epsilon+\delta}(X_{\text{test}}, Y_{\text{test}}) \leq q_\alpha \mid X_{\text{test}} \in K\right) + \mathbb{P}(X_{\text{test}} \notin K) \\
&\overset{\text{by (22)}}{\leq} \mathbb{P}(X_{\text{test}} \in K) \cdot \underbrace{\mathbb{P}\left(\underline{s}_\epsilon(X_{\text{test}}, Y_{\text{test}}) \leq q_\alpha + \rho \mid X_{\text{test}} \in K\right)}_{=F_\epsilon(q_\alpha+\rho) \leq F_\epsilon(q_\alpha) + \frac{\eta}{2}} + \frac{\eta}{2} \\
&\leq \mathbb{P}(X_{\text{test}} \in K) \cdot \left[\mathbb{P}\left(\underline{s}_\epsilon(X_{\text{test}}, Y_{\text{test}}) \leq q_\alpha \mid X_{\text{test}} \in K\right) + \frac{\eta}{2}\right] + \frac{\eta}{2} \\
&\leq \mathbb{P}\left(\underline{s}_\epsilon(X_{\text{test}}, Y_{\text{test}}) \leq q_\alpha\right) + \eta \\
&= \overline{\gamma}(\epsilon) + \eta
\end{aligned}$$

This entails $|\overline{\gamma}(\epsilon') - \overline{\gamma}(\epsilon)| \leq \eta$ for all $\epsilon' \in [\epsilon, \epsilon + \delta]$, and proves that $\overline{\gamma}$ is right-continuous.

The proof that $\underline{\gamma}$ is left-continuous follows from similar arguments. Put briefly: for all $\epsilon > 0$ and $\eta > 0$, there exist a

compact subset $K \subset \mathcal{X}$ and two real numbers $\rho > 0$ and $\delta \in (0, \epsilon)$ such that

$$
\begin{aligned}
\underline{\gamma}(\epsilon - \delta) &= \mathbb{P}\left(\bar{s}_{\epsilon - \delta}\left(X_{\text{test}}, Y_{\text{test}}\right) \leq q_\alpha\right) \\
&\leq \mathbb{P}\left(X_{\text{test}} \in K\right) \cdot \mathbb{P}\left(\bar{s}_{\epsilon - \delta}\left(X_{\text{test}}, Y_{\text{test}}\right) \leq q_\alpha \mid X_{\text{test}} \in K\right) + \mathbb{P}\left(X_{\text{test}} \notin K\right) \\
&\overset{s(\cdot, y) \text{ u.c. on } K}{\leq} \mathbb{P}\left(X_{\text{test}} \in K\right) \cdot \mathbb{P}\left(\bar{s}_\epsilon\left(X_{\text{test}}, Y_{\text{test}}\right) \leq q_\alpha + \rho \mid X_{\text{test}} \in K\right) + \frac{\eta}{2} \\
&\leq \mathbb{P}\left(X_{\text{test}} \in K\right) \cdot \left[\mathbb{P}\left(\bar{s}_\epsilon\left(X_{\text{test}}, Y_{\text{test}}\right) \leq q_\alpha \mid X_{\text{test}} \in K\right) + \frac{\eta}{2}\right] + \frac{\eta}{2} \\
&\leq \mathbb{P}\left(\bar{s}_\epsilon\left(X_{\text{test}}, Y_{\text{test}}\right) \leq q_\alpha\right) + \eta \\
&= \underline{\gamma}(\epsilon) + \eta
\end{aligned}
$$

$\square$

**N.B.** The proof is a little easier if $s(\cdot, y)$ is uniformly continuous on $\mathcal{X}$ (e.g. if $\mathcal{X}$ is compact or $s(\cdot, y)$ is Lipschitz). In that case, there is no need to work conditionally on $\{X_{\text{test}} \in K\}$.

### B.2. Concentration of $\overline{\gamma}_m$ around $\overline{\gamma}$

Let $\mathcal{D}_{\text{eval}} = (X_i, Y_i)_{1 \leq i \leq m}$ be $m \geq 2$ independent copies of $(X_{\text{test}}, Y_{\text{test}})$.

For $\epsilon > 0$, let

$$
\overline{\gamma}_m(\epsilon) := \frac{1}{n} \sum_{i=1}^m \mathbb{1}_{\underline{s}_\epsilon(X_i, Y_i) \leq q_\alpha}.
$$

Let $\delta \in (0, 1)$. Since $m\overline{\gamma}_m(\epsilon) \sim \text{Binomial}(m, \overline{\gamma}(\epsilon))$, the estimator

$$
\overline{\gamma}_m^+(\epsilon, \delta) := \max\left\{p \in [0, 1] : \text{Bin}_{m,p}(m\overline{\gamma}_m(\epsilon)) \geq \delta\right\},
$$

where $\text{Bin}_{m,p}$ is the c.d.f. of $\text{Binomial}(m, p)$. This estimator satisfies (e.g., Theorem 3.3 by Langford & Schapire 2005), for all $\epsilon \geq 0$ and $\delta \in (0, 1)$,

$$
\mathbb{P}\left(\overline{\gamma}(\epsilon) \leq \overline{\gamma}_m^+(\epsilon, \delta)\right) \geq 1 - \delta \tag{23}
$$

Applying (23) $m - 1$ times with properly chosen values of $\epsilon$ (of a quantile flavor), and using the fact that both $\overline{\gamma}$ and $\overline{\gamma}_m^+(\cdot, \delta)$ are a.s. right-continuous and non-decreasing, we obtain the following "uniform" concentration inequality.

**Theorem B.2.** *Assume that A1 and A2 from 3.2 hold true. Let $(X_i, Y_i)_{1 \leq i \leq m}$ be $m \geq 2$ independent copies of $(X_{\text{test}}, Y_{\text{test}})$. Then:*

$$
\mathbb{P}\left(\forall \epsilon \geq 0, \overline{\gamma}(\epsilon) \leq \overline{\gamma}_m^+(\epsilon, \frac{\delta}{m-1}) + \frac{1}{m}\right) \geq 1 - \delta.
$$

As seen in the proof below, $\overline{\gamma}$ and $\overline{\gamma}_m^+(\cdot, \delta)$ are right-continuous a.s., so that the above probability is well defined (the $\forall \epsilon$ can be replaced with a countable $\forall \epsilon_k$).

*Proof.* We proceed in three steps.

**Step 1: right-continuity of $\overline{\gamma}$ and $\overline{\gamma}_m^+(\cdot, \delta)$**

- the fact that $\overline{\gamma}$ is right-continuous follows from Subsection B.1.

- likewise, using this result with the empirical distribution $\frac{1}{m} \sum_{i=1}^m \delta_{(X_i, Y_i)}$, we can see that the random function $\epsilon \mapsto \overline{\gamma}_m(\epsilon)$ is right-continuous, and thus locally constant to the right of any $\epsilon \geq 0$. This implies that $\epsilon \mapsto \overline{\gamma}_m^+(\epsilon, \delta)$ is also right-continuous on $[0, +\infty)$.

**Step 2: pointwise concentration and union bound** For every $k \in \{1, \cdots, m-1\}$, we set $\epsilon_k := \inf \left\{ \epsilon \geq 0 : \overline{\gamma}(\epsilon) \geq \frac{k}{m} \right\}$, with the convention that $\inf \varnothing = +\infty$. Let also $\epsilon_m := +\infty$. Note that $0 \leq \epsilon_1 \leq \epsilon_2 \leq \cdots \leq \epsilon_m \leq +\infty$ and that $\overline{\gamma}(\epsilon_k) \geq \frac{k}{m}$ whenever $\epsilon_k < \infty$ (by right-continuity of $\overline{\gamma}$ on $[0, +\infty)$).

First note that, if $\epsilon_1 = +\infty$, then $\overline{\gamma}(\epsilon) < \frac{1}{m}$ for all $\epsilon \geq 0$ by def of $\epsilon_1$). In this case, the conclusion of Theorem B.2 would hold trivially.

We can thus assume without loss of generality that $\epsilon_1 < +\infty$, and set

$$K := \max \left\{ k \in \{1, \cdots, m-1\} : \epsilon_k < +\infty \right\}.$$

Note that $0 \leq \epsilon_1 \leq \cdots \leq \epsilon_K < \epsilon_{K+1} = \cdots = \epsilon_m = +\infty$.

Combining (23) with a union bound, and using $K \leq m-1$, we get:

$$\mathbb{P}\left( \forall k \in \{1, \cdots, K\}, \overline{\gamma}(\epsilon_k) \leq \overline{\gamma}_m^+\left(\epsilon_k, \frac{\delta}{m-1}\right) \right) \geq 1 - \frac{K\delta}{m-1} \geq 1 - \delta. \tag{24}$$

**Step 3: bridging the gaps** Let $\Omega_\delta := \left\{ \forall k \in \{1, \cdots, K\}, \overline{\gamma}(\epsilon_k) \leq \overline{\gamma}_m^+(\epsilon_k, \frac{\delta}{m-1}) \right\}$ be the event appearing in (24). We now work on the event $\Omega_\delta$.

Let $\epsilon \geq \epsilon_1$. Note that $\epsilon$ belongs to one of the following K disjoint (possibly empty) intervals:

$$I_k = [\epsilon_k, \epsilon_{k+1}), k \in \{1, \cdots, K\} \quad \text{(by convention, } I_k = \varnothing \text{ if } \epsilon_k = \epsilon_{k+1}).$$

We let $k \in \{1, \cdots, K\}$ be such that $\epsilon \in I_k$, i.e., $\epsilon_k \leq \epsilon < \epsilon_{k+1}$.

Recall that $\overline{\gamma}(\epsilon_k) \geq \frac{k}{m}$, and note that $\overline{\gamma}(\epsilon) \leq \frac{k+1}{m}$.

Therefore,

$$
\begin{aligned}
\overline{\gamma}(\epsilon) &\leq \frac{k+1}{m} \\
&\leq \overline{\gamma}(\epsilon_k) + \frac{1}{m} \\
&\leq \overline{\gamma}_m^+(\epsilon_k, \frac{\delta}{m-1}) + \frac{1}{m} && \text{(we work on } \Omega_\delta) \\
&\leq \overline{\gamma}_m^+(\epsilon, \frac{\delta}{m-1}) + \frac{1}{m} && \text{(by } \epsilon_k \leq \epsilon \text{ and monotonicity of } \overline{\gamma}_m^+(\cdot, \frac{\delta}{m-1}))
\end{aligned}
$$

Since we also have $\overline{\gamma}(\epsilon) \leq \frac{1}{m}$ if $\epsilon_1 > 0$ and $\epsilon \in [0, \epsilon_1)$ (by definition of $\epsilon_1$), we just proved that, on the event $\Omega_\delta$,

$$\forall \epsilon \geq 0, \overline{\gamma}(\epsilon) \leq \overline{\gamma}_m^+(\epsilon, \frac{\delta}{m-1}) + \frac{1}{m}.$$

Recalling that $\mathbb{P}(\Omega_\delta) \geq 1 - \delta$ concludes the proof. $\qquad \square$

We can control $\underline{\gamma} = \mathbb{P}(\overline{s}_\epsilon(X_{\text{test}}, Y_{\text{test}}) \leq q_\alpha)$ similarly (up to a coin flip), using

$$\underline{\gamma}_m := \frac{1}{m} \sum_{i=1}^m \mathbb{1}_{\overline{s}_\epsilon(X_i, Y_i) \leq q_\alpha}$$

and

$$\underline{\gamma}_m^-(\epsilon, \delta) := 1 - \max \left\{ p \in [0, 1] : \text{Bin}_{m,p}(m(1 - \underline{\gamma}_m(\epsilon))) \geq \delta \right\}.$$

We have the following lower bound.

**Theorem B.3.** *Assume that A1 and A2 from 3.2 hold true. Let $(X_i, Y_i)_{1 \leq i \leq m}$ be $m \geq 2$ independent copies of $(X_{\text{test}}, Y_{\text{test}})$. Then:*

$$\mathbb{P}\left( \forall \epsilon > 0, \underline{\gamma}(\epsilon) \geq \underline{\gamma}_m^-\left(\epsilon, \frac{\delta}{m-1}\right) - \frac{1}{m} \right) \geq 1 - \delta. \tag{25}$$

First, note that our high probability lower bound on $\underline{\gamma}(\epsilon)$ is similar to our high probability upper bound on $\overline{\gamma}(\epsilon)$, by a simple coin flip. Indeed, note that:

$$1 - \underline{\gamma}(\epsilon) = \mathbb{P}\left(\bar{s}_\epsilon(X_{\text{test}}, Y_{\text{test}}) > q_\alpha\right).$$

$$1 - \underline{\gamma}_m(\epsilon) := \frac{1}{m} \sum_{i=1}^{m} \mathbb{1}_{\bar{s}_\epsilon(X_i, Y_i) > q_\alpha} \tag{26}$$

$$1 - \underline{\gamma}_m^-(\epsilon, \delta) := \max\{p \in [0, 1] : \text{Bin}_{m,p}\left(m \cdot (1 - \underline{\gamma}(\epsilon))\right) \geq \delta\}$$

**Issue**   Though these functions are non-decreasing in $\epsilon$, they are *left*-continuous. This calls for additional technicalities, as seen in the proof below.

*Proof.* We also proceed with three steps: many arguments are identical, but not all.

**Step 1: left continuity of $\underline{\gamma}$ and $\underline{\gamma}_m^-(\cdot, \delta)$ on $(0, +\infty)$**   This follows from similar arguments as in the proof of Theorem B.2.

**Step 2: Pointwise concentration and union bound**   For every $k \in \{1, \cdots, m-1\}$, we set $\epsilon_k := \inf\{\epsilon \geq 0 : 1 - \underline{\gamma}(\epsilon) \geq \frac{k}{m}\}$, with the convention that $\inf \varnothing = +\infty$. Let also $\epsilon_m := +\infty$.

Note that $0 \leq \epsilon_1 \leq \epsilon_2 \leq \cdots \leq \epsilon_m \leq +\infty$ and that $1 - \underline{\gamma}(\epsilon_k) \leq \frac{k}{m}$ whenever $0 < \epsilon_k \leq +\infty$ (by left continuity of $\underline{\gamma}$ on $(0, +\infty)$).

First note that, if $\epsilon_1 = +\infty$, then $\underline{\gamma}(\epsilon) > 1 - \frac{1}{m}$ for all $\epsilon \geq 0$ (by def of $\epsilon_1$). In this case, the conclusion of Theorem B.3 would hold trivially.

We can thus assume without loss of generality that $\epsilon_1 < +\infty$, and set

$$K := \max\left\{k \in \{1, \cdots, m-1\} : \epsilon_k < +\infty\right\}.$$

Note that $0 \leq \epsilon_1 \leq \cdots \leq \epsilon_K \leq \epsilon_{K+1} = \cdots = \epsilon_m = +\infty$.

We now need additional technicalities.

For any $q \in \mathbb{N}^* = \{1, 2, \cdots\}$ and $k \in \{1, \cdots, K\}$, we set

$$\epsilon_k^q = \begin{cases} \left(1 - \frac{1}{q}\right)\epsilon_k + \frac{1}{q}\epsilon_{k+1} & \text{if } k \leq K-1 \\ \epsilon_k + \frac{1}{q} & \text{if } k = K \end{cases}$$

Note that

$$\begin{cases} \epsilon_k < \epsilon_k^q < \epsilon_{k+1} & \text{if } \epsilon_k < \epsilon_{k+1}. \\ \epsilon_k = \epsilon_k^q = \epsilon_{k+1} & \text{if } \epsilon_k = \epsilon_{k+1}. \end{cases}$$

and that $\epsilon_k^{q+1} \leq \epsilon_k^q$.

By Langford (2005, Theorem 3.3), we have similarly to (23),

$$\forall \epsilon \geq 0, \forall \delta \in (0, 1), \mathbb{P}\left(\underline{\gamma}(\epsilon) \geq \underline{\gamma}_m^-(\epsilon, \delta)\right) \geq 1 - \delta. \tag{27}$$

Let $\delta \in (0, 1)$. Combining (27) with a union bound, we get, for any $q \geq 1$,

$$\mathbb{P}\left(\forall k \in \{1, \cdots, K\}, \underline{\gamma}(\epsilon_k^q) \geq \underline{\gamma}_m^-(\epsilon_k^q, \frac{\delta}{m-1})\right) \geq 1 - \delta. \tag{28}$$

We now work on the event

$$\Omega_\delta^q := \left\{\forall k \in \{1, \cdots, K\}, \underline{\gamma}(\epsilon_k^q) \geq \underline{\gamma}_m^-(\epsilon_k^q, \frac{\delta}{m-1})\right\}.$$

We set $\mathcal{E}^q := \left(\bigcup_{k=1}^{K-1}(\epsilon_q^k, \epsilon_{k+1}]\right) \cup (\epsilon_K^q, +\infty)$, where $(\epsilon_k^q, \epsilon_{k+1}] = \varnothing$ if $\epsilon_k = \epsilon_{k+1}$. Note that $\mathcal{E}^q \subseteq \mathcal{E}^{q+1} \subseteq (\epsilon_1, +\infty)$ and $\bigcup_{q \geq 1} \mathcal{E}^q = (\epsilon_1, +\infty)$

Now, let $\epsilon \in \mathcal{E}^q$. There exists $k \in \{1, \cdots, K\}$ such that $k \leq K - 1$ and $\epsilon \in (\epsilon_k^q, \epsilon_{k+1}]$, or $k = K$ and $\epsilon \in (\epsilon_K^q, +\infty)$. In particular, $\epsilon_k < \epsilon_k^q$ whatever the value of k.

Note that, if $k \leq K - 1$, we have $0 < \epsilon_{k+1} < +\infty$ and therefore $1 - \underline{\gamma}(\epsilon_{k+1}) \leq \frac{k+1}{m}$. This entails (using $\epsilon \leq \epsilon_{k+1}$ and the fact that $1 - \underline{\gamma}$ is non-decreasing)

$$1 - \underline{\gamma}(\epsilon) \leq \frac{k+1}{m},$$

which is also true $k = K$ (by def of $\epsilon_{K+1}$ if $K + 1 \leq m - 1$, or trivial if $K = m = 1$). Therefore, whatever the value of $k$ above,

$$
\begin{aligned}
1 - \underline{\gamma}(\epsilon) &\leq \frac{k+1}{m} \\
&\leq 1 - \underline{\gamma}(\epsilon_k^q) + \frac{1}{m} && \left(\text{since } 1 - \underline{\gamma}(\epsilon_k^q) \geq \frac{k}{m}, \text{ by def of } \epsilon_k \text{ and } \epsilon_k^q > \epsilon_k\right) \\
&\leq 1 - \underline{\gamma}_m^-(\epsilon_k^q, \frac{\delta}{m-1}) + \frac{1}{m} && \left(\text{we work on } \Omega_\delta^q\right) \\
&\leq 1 - \underline{\gamma}_m^-(\epsilon, \frac{\delta}{m-1}) + \frac{1}{m} && \left(\text{by } \epsilon_k^q \leq \epsilon \text{ and monotonicity of } \underline{\gamma}_m^-(\cdot, \frac{\delta}{m-1})\right)
\end{aligned}
\tag{29}
$$

Rearranging terms, we proved that, on the event $\Omega_\delta^q$,

$$\forall \epsilon \in \mathcal{E}^q, \underline{\gamma}(\epsilon) \geq \underline{\gamma}_m^-(\epsilon, \frac{\delta}{m-1}) - \frac{1}{m}. \tag{30}$$

We now let $q \to +\infty$. More precisely, recalling that $(\epsilon_1, +\infty) = \cup_{q \geq 1} \mathcal{E}^q$,

$$
\begin{aligned}
&\mathbb{P}\left(\forall \epsilon \in (\epsilon_1, +\infty), \underline{\gamma}(\epsilon) \geq \underline{\gamma}_m^-(\epsilon, \frac{\delta}{m-1}) - \frac{1}{m}\right) \\
&= \mathbb{P}\left(\cap_{q \geq 1}\{\forall \epsilon \in \mathcal{E}^q, \underline{\gamma}(\epsilon) \geq \underline{\gamma}_m^-(\epsilon, \frac{\delta}{m-1}) - \frac{1}{m}\}\right) \\
&= \lim_{q \to +\infty} \mathbb{P}\left(\forall \epsilon \in \mathcal{E}^q, \underline{\gamma}(\epsilon) \geq \underline{\gamma}_m^-(\epsilon, \frac{\delta}{m-1}) - \frac{1}{m}\right) && \left(\text{since } \mathcal{E}^q \subseteq \mathcal{E}^{q+1}\right) \\
&\geq 1 - \delta. && \left(\text{since } \mathbb{P}(\Omega_\delta^q) \geq 1 - \delta\right)
\end{aligned}
$$

Nothing that, if $\epsilon_1 > 0$, then $1 - \underline{\gamma}(\epsilon) \leq \frac{1}{m}$ for all $\epsilon \in [0, \epsilon_1]$, concludes the proof. $\square$

## C. On Measurability Details

In this short section, we discuss minor mathematical details that are yet important to ensure that all probabilities under study are well defined. In short, probabilities should be studied for sets that are *measurable*, which can fail in general when manipulating, e.g., continuously-infinite suprema or infima of measurable functions.

We start by describing two measurability issues in items 1 and 2 below. We then explain that, if the assumptions of the paper hold true, then we work with continuous and thus measurable functions, so that these issues do not arise (see Proposition C.1).

1. The modified scores $\bar{s}_\epsilon(x, y) := \sup_{\tilde{x} \in \mathcal{B}_\epsilon(x)} s(\tilde{x}, y)$ and $\underline{s}_\epsilon(x, y) := \inf_{\tilde{x} \in \mathcal{B}_\epsilon(x)} s(\tilde{x}, y)$ might not be measurable functions of $(x, y)$. This condition is required so that the sets $\{Y \in \bar{C}_{\alpha,\epsilon}(X)\} = \{\underline{s}_\epsilon(X, Y) \leq q_\alpha\}$ and $\{Y \in \underline{C}_{\alpha,\epsilon}(X)\} = \{\bar{s}_\epsilon(X, Y) \leq q_\alpha\}$ are measurable (i.e., well defined events).

2. Even so, the function $(x, y) \mapsto \sup_{\tilde{x} \in \mathcal{B}_\epsilon(x)} \underline{s}_\epsilon(\tilde{x}, y)$ might not be measurable, which is required for the quantity $\mathbb{P}\left(\forall \tilde{x} \in \mathcal{B}_\epsilon(X), Y \in \bar{C}_{\alpha,\epsilon}(\tilde{x})\right)$ to be well defined.

**Recalling Assumptions 3.2**

A1 $\mathcal{X}$ is a convex and closed subset of $\mathbb{R}^d$

A2 $s(\cdot, y)$ is continuous for all $y \in \mathcal{Y}$

**Proposition C.1.** *Let $\epsilon \geq 0$ and assume that A1 and A2 from Assumption 3.2 hold true. Then the functions $\underline{s}_\epsilon(\cdot, y)$, $\bar{s}_\epsilon(\cdot, y)$, and $x \mapsto \sup\limits_{\tilde{x} \in \mathcal{B}_\epsilon(x)} \underline{s}_\epsilon(\tilde{x}, y)$ are continuous for all $y \in \mathcal{Y}$.*

The proof follows from uniform continuity arguments that are very similar to those used in the proof of Proposition B.1, and is thus omitted.

# D. Extending VRCP

In this section, we explictly develop extensions of formal verification methods for robust conformal prediction. First, we recall the following robustness regimes:

- **Robust conformal prediction set** Here, we return a conformal prediction set that verifies user-specified conformal coverage under adversarial perturbations. However, this certificate can yield overly conservative results on clean data resulting in increased conformal prediction set sizes.

- **Certifiably robust vanilla conformal prediction** Through estimations of the conservative and restrictive prediction sets on an extra set of holdout data $\mathcal{D}_{\text{eval}}$, our theoretical results ensure that for a given adversarial attack budget, our conformal coverage on adversarially perturbed data will remain within a range of $[\underline{\gamma}_m^-, \overline{\gamma}_m^+]$ with high probability.

In *VRCP* (Jeary et al., 2024), the authors develop a method allowing the computation of supersets of the conservative prediction set around a point $x$ leveraging formal verification methods. To this end, they prove that the conservative prediction set verifies robust coverage bounds under adversarial attacks. In order to extend verification methods to the computation of the restrictive prediction set (necessary for the coverage bounds on vanilla CP under attack) and the computation of a lower bound for the robustness radius of a vanilla prediction set, we give the following insights.

**Restrictive prediction set** Given that formal verification methods often return the upper and lower bounds of every logit of the output layer. This facilitates the computation of a subset of the restrictive prediction set with formal verification methods.

# E. About Score Functions: Sigmoid vs Softmax

## E.1. Vanilla CP Setting

We recall the following score definitions:

$$
\begin{aligned}
s_{\text{LAC Softmax}}(x, y) &= 1 - \text{softmax}\left(\frac{f(x)}{T}\right)_y \\
s_{\text{LAC Sigmoid}}(x, y) &= 1 - \text{sigmoid}\left(\frac{f(x)_y - b}{T}\right)
\end{aligned}
\tag{31}
$$

To evaluate the impact of using our LAC Sigmoid score as compared to a classic LAC Softmax score function, we provide the following empirical results for vanilla CP on the *CIFAR-10* dataset with a ResNet50 model with $\alpha = 0.1$:

| Method | Coverage (%) | Set size |
|--------|--------------|----------|
| LAC Softmax | 90.04 | 1.088 |
| LAC Sigmoid | 90.28 | 1.144 |

Table 4: Comparison of conformal score performance LAC Softmax vs LAC Sigmoid.

This ablation study results in marginally bigger vanilla CP set sizes for LAC Sigmoid scores compared to softmax ones with scaled temperature.

### E.2. Robust CP Setting

Using the 1-Lipschitz nature of our models, we can also use exact worst-case LAC Sigmoid and Softmax score computations under adversarial logit perturbations. Instead of relying on Eq. 19, we can leverage the monotonicity of the sigmoid function to compute tighter bounds, e.g:

$$1 - \text{sigmoid}\left(\frac{f(\tilde{x})_y + L_n \times \epsilon - b}{T}\right) \leq \min_{x \in \mathcal{B}_\epsilon(\tilde{x})} s_{\text{LAC Sigmoid}}(x, y) \tag{32}$$

where $T$ and $b$ are the temperature and the bias of the score. This inequality yields tighter bounds than those based solely on global Lipschitz constants, with no extra cost.

Similarly, given that the softmax function exhibits monotonicity in each of its parameters (ceteris paribus), its maximum and minimum values within a hyper-rectangular region are attained at the corners of that region, i.e.,

$$1 - \text{softmax}\begin{pmatrix} (f(\tilde{x})_0 - L_n \times \epsilon)/T \\ \vdots \\ (f(\tilde{x})_y + L_n \times \epsilon)/T \\ \vdots \\ (f(\tilde{x})_{c-1} - L_n \times \epsilon)/T \end{pmatrix}_y \leq \min_{x \in \mathcal{B}_\epsilon(\tilde{x})} s_{\text{LAC Softmax}}(x, y) \tag{33}$$

where $T$ is the temperature scaling factor of the softmax.

As shown below, for a 1-Lipschitz VGG with 25M parameters on *CIFAR-10* with $\epsilon = 0.03$ and $\alpha = 0.1$, LAC Sigmoid demonstrates considerably smaller robust prediction sets.

| Method | Coverage (%) | Set size |
|---|---|---|
| LAC Softmax | 95.6 | 2.38 |
| LAC Sigmoid | 94.9 | 1.72 |

Table 5: Robust CP performance for different conformal scores on *CIFAR-10*. Under the same conditions as Fig 3.

Although the LAC Sigmoid score is marginally worse in vanilla CP conditions, the superior performances obtained in Table 5 motivates its use in the paper.

## F. Model Implementation

### F.1. Training Time Overhead

In this section we describe the computational overhead of our implementation on a *CIFAR-10* shaped set of random data. In our experimental setup, we use a standard neural network with two convolutional layers followed by max pooling operations and a linear layer. We study how a drop-in replacement of the vanilla PyTorch layers by our chosen Lipschitz constrained layers affects the overall training time of our network. Importantly, our implementation is batch-size independent since it only relies on the values of the weights. Therefore the overall cost of training Lipschitz constrained networks is mitigated when using large batch sizes.

| Batch size | Train time (%) |
|---|---|
| 1024 | 20.31 |
| 2048 | 10.26 |

Table 6: Training time on random data of shape $(32, 32, 3)$ when dropping in the Lipschitz-constrained layers. For reference, our ResNeXt model of Figure 3 introduces a 9.6% runtime overhead on TinyImageNet compared to an identical unconstrained model.

For more information regarding the computational overhead of Lipschitz constrained architectures w.r.t standard unconstrained architectures, we refer the reader to the comprehensive benchmark of (Boissin et al., 2025).

**Improvements**   In our paper we use a relatively standard implementation of Lipschitz parametrized networks with an orthogonality constraint. Many recent works have focused on providing efficient and expressive Lipschitz network implementations (Araujo et al., 2023; Leino et al., 2021; Meunier et al., 2022) that could be applied in the context of our study.

### F.2. Placement of Method

**Overall placement of method**   See Table 3 for an overall comparison of certifiably robust methods for conformal prediction.

**Hyperparameters of all methods**   We find that other methods have the following hyperparameters :

| Method | Hyperparameters |
|--------|-----------------|
| *RSCP* | Smoothing strength $\sigma$, Monte Carlo samplings $n_{mc}$. |
| *RSCP+* | Smoothing strength $\sigma$, Monte Carlo samplings $n_{mc}$, confidence of smoothing $\beta$. |
| + PTT | Temperature $T$, bias $b$, number of holdout points $n_{\text{holdout}}$. |
| + RCT | Temperature $\tau^{\text{soft}}$, regularization factors $\lambda$ and $\kappa$. |
| *aPRCP* | Conservativeness hyperparameter $s$. |
| *CAS* | Smoothing strength $\sigma$, Monte Carlo samplings $n_{mc}$. |
| *VRCP* | Choice of formal verification method (counts as one hyperparameter) plus any score specific parameters. |
| *lip-rcp* | Score temperature $T$ and bias $b$. |

Table 7: Hyperparameters for each certified-robustness method

## G. Experimental Settings

Robust CP methods are heavily dependent on the quality and compatibility of the underlying model. Therefore, we use separate neural networks to ensure a fair evaluation of robust CP methods. In the following sections, we define the models we used for benchmarking both robust CP and worst-case vanilla CP coverage bounds.

### G.1. System Requirements

All experiments were conducted on a system equipped with two NVIDIA GeForce RTX 4090 GPUs, each providing 24 GB of GDDR6X memory.

### G.2. VRCP Models

Given that formal verification methods have an inherent trade-off between computational overhead and tightness of their estimations, we use specifically tailored models from Appendix B of Jeary et al. (2024). We then compare these models to Lipschitz by design models in the experiments of § 5.1 and § 5.2.

Importantly, using larger and deeper models with verification methods is not only computationally punitive, it also usually tends to worsen the obtained bounds as they become looser and looser.

### G.3. Smoothing Models

We use a ResNet50 architecture in the experiments we run using *CAS* as it offers a good baseline for robust CP results as done in (Zargarbashi et al., 2024). Given the already costly nature of smoothing methods at inference, we follow a standard scheme at training time to avoid additional computational burden. An exception stands for the `CIFAR-10` dataset: models trained with Gaussian noise as per the procedure of Salman et al. (2019) still fail in low MC iteration settings. Indeed, we test a ResNet110 model trained with noise augmentation from the aforementioned procedure: yet it still exhibits uninformative set sizes of 10 for $n_{\text{mc}} = 64$ on every point of the test set. Moreover, we use the recommended $\sigma$ value of $\sigma = 2 \cdot \epsilon$ for every smoothing experiment conducted.

### G.4. Lipschitz Models

The architecture we use for our experiments on *CIFAR-10*, *CIFAR-100* and *TinyImageNet* datasets follows a ResNeXt-like design, with an initial convolutional stem followed by four stages of increasing feature channels ($64 \rightarrow 128 \rightarrow 256 \rightarrow 512$).

Each stage comprises grouped-convolution residual blocks, with downsampling applied between stages to reduce spatial dimensions. The network then employs global average pooling, a fully connected bottleneck, and a classification layer.

Regarding our experiments on the *ImageNet* dataset, we leverage a more efficient implementation of Lipschitz parametrized networks, as described in (Boissin et al., 2025). This implementation can be found in the `orthogonium` library.

Regarding our *ImageNet* model we use the following architecture. Starting with two strided convolutions (7×7 → 3×3 kernels) that reduce spatial resolution while expanding channels to 256. Four subsequent processing stages progressively double channel dimensions (256 → 512 → 1024 → 2048), each containing three residual blocks with depthwise 5×5 convolutions. Blocks employ channel expansion/contraction (2 × width), GroupSort2 activations, and learned residual mixing that respects 1-Lipschitz constraints. Then, the spatial features are condensed via global average pooling (7×7 window) before final classification by a linear layer that enforces the 1-Lipschitz condition on each output.

**Training objectives**    Our neural networks are trained with Optimal Transport inspired objectives such as the Hinge Kantorovich-Rubinstein loss function introduced in (Serrurier et al., 2021). These loss functions are particularly effective to enforce model robustness.

### G.5. Training Hyperparameters

We train our Lipschitz neural networks with the AdamW optimizer with a learning rate 1e-3. Also, we use the `SoftHKRMulticlassLoss` from the `deel-torchlip` library with the following standard values:

| Dataset | Margin | Temperature | Epochs | Alpha |
|---|---|---|---|---|
| *CIFAR-10* | 0.6 | 5.0 | 130 | 0.975 |
| *CIFAR-100* | 0.6 | 5.0 | 220 | 0.975 |
| *TinyImageNet* | 0.3 | 5.0 | 80 | 0.975 |

Table 8: Standard training hyperparameter values for the loss function on the different datasets.

## H. Calibration Poisoning Results

**Calibration time label flipping attacks**    Most conformal score functions are often not Lipschitz continuous with respect to $y \in \mathcal{Y}$. Therefore we devise no straightforward certificate for label flipping attacks. However, the defense to label flipping attacks defined in (Zargarbashi et al., 2024) that leverages the computation of a maximum quantile shift when scores are permuted is valid for our networks too.

**Calibration time feature poisoning attacks**    In the context of calibration-time feature poisoning however the Lipschitz condition of our score with respect to $x \in \mathcal{X}$ becomes greatly advantageous. Indeed, the problem of computing the maximum quantile shift becomes simplified in the sense that every score verifies a $L_s$ Lipschitz condition and is interdependent from the scores of other calibration samples for the true class. We devise a simple Linear Programming script described in Figure 6 to handle the computation of the maximum and minimum quantile shift under attack of budget $(k, \epsilon)$. We verify our Linear Programming solution empirically on the *CIFAR-10* dataset (see Table 9).

**NB**    Importantly, our algorithm assumes that the quantile computation done during the calibration step returns $q_\alpha = \overrightarrow{R}_{\lceil (n+1)(1-\alpha) \rceil}$.

| $\epsilon$ | k | Metric | Mean | Std Dev |
|---|---|---|---|---|
| 0 | – | Val. accuracy | 0.7260 | – |
| | | Set size | 2.0690 | 0.0387 |
| 0.25 | 6 | Robust CP $\gamma$ | 0.8982 | 0.0085 |
| | | Robust CP set size | 2.0699 | 0.0467 |
| 0.25 | 50 | Robust CP $\gamma$ | 0.9090 | 0.0068 |
| | | Robust CP set size | 2.1872 | 0.0466 |

Table 9: Summary of empirical coverage and prediction set sizes across different attack budgets $(k, \epsilon)$ for feature poisoning on the calibration set ($n_{cal} = 4750$). Each experiment was repeated 10 times with a randomly selected calibration subset.

```python
def compute_robust_threshold(scores, k, epsilon, alpha, min_value: float = 0):
    n_cal = len(scores) + 1
    idxs_sort = np.argsort(scores)
    idx_threshold = np.floor(alpha * n_cal).astype(int)
    sorted_scores = np.sort(scores)

    seen = []
    remaining_k = k

    while remaining_k >= 0:
        if idxs_sort[idx_threshold] in seen:
            scores[idxs_sort[idx_threshold + 1]] -= epsilon
            return np.sort(scores)[idx_threshold]
        seen.append(idxs_sort[idx_threshold])

        scores[idxs_sort[idx_threshold]] -= epsilon
        idxs_sort = idxs_sort[np.argsort(scores[idxs_sort])]
        remaining_k -= 1

    return np.sort(scores)[idx_threshold]
```

Figure 6: Our function for computing the maximum quantile shift and therefore certifying the robustness of prediction sets under calibration time feature poisoning attacks.

## I. About the Tightness of Score Bounds

To estimate how tight our score estimations are we evaluate all our methods on the same model. We train a VGG-like 1-LipNet with 10.7M parameters from Boissin et al. (2025) on CIFAR-10. Then, we evaluate robust CP methods (average across 10 runs).

| Method | Coverage (%) | Set size | Runtime (s) |
|---|---|---|---|
| *CAS* $n_{\mathrm{mc}} = 1024$ | 94.60 | 2.302 | 2615 |
| *lip-rcp* | 92.83 | 1.889 | 10 |
| *VRCP-I/C* (CROWN) | OOM | OOM | N/A |

Table 10: Average robust CP coverage, set size, and runtime (per run) with $\epsilon = 0.03$ across 10 runs.

CROWN does not scale to such a deep network due to its inner complexity. Additionally, given the cost of the CAS method, we did not tune the $\sigma$ smoothing hyperparameter set it to $\sigma = 2.\epsilon$ as recommended in previous works (Yan et al., 2024). Optimizing this hyperparameter could lead to better results as obtained in (Zargarbashi & Bojchevski, 2025).

