# OpenReview forum: "Efficient Robust Conformal Prediction via Lipschitz-Bounded Networks"
_ICML.cc/2025/Conference — ICML 2025 poster_

### Official Review · Reviewer_XbCa · 2025-02-28

**Overall Recommendation:** 4

**Summary:**

This paper explores how to leverage LipNet and a novel robust conformal score algorithm for robust prediction. Previously proposed robust conformal prediction methods each have their own limitations, such as high computational complexity, making them difficult to scale to large datasets like ImageNet. By utilizing the properties of LipNet, this paper introduces a simple conformal score estimation method with lower computational complexity. The effectiveness of the proposed approach is validated on large-scale datasets, including CIFAR and ImageNet, in the latter part of the paper.

**Claims And Evidence:**

Yes

**Essential References Not Discussed:**

N/A

**Experimental Designs Or Analyses:**

Yes

**Methods And Evaluation Criteria:**

Yes

**Other Comments Or Suggestions:**

This paper is clear and well-structured, with substantial theoretical contributions and a comprehensive set of experiments. The proposed method has the potential to attract significant interest from various fields, such as conformal prediction and Lipschitz neural networks. Hence, I am inclined to accept it.

## update after rebuttal: The authors have addressed my concerns, so I recommend acceptance.

**Other Strengths And Weaknesses:**

Strengths:
1. The paper is clearly written, with well-designed illustrations and elegant notation. Even readers unfamiliar with LipNet and conformal prediction can quickly grasp the key ideas presented.
2. This paper contributes to two fields. In the domain of conformal prediction, it introduces a new perspective of robustness and proposes a new approach to enhance the robustness of vanilla CP. In the field of Lipschitz neural networks, it leverages existing LipNet techniques to estimate the robust CP score.
3. The experiments in this paper are conducted on various real-world datasets, ensuring a high level of reliability.

Weaknesses/suggestions: Based on my knowledge, this paper does not exhibit any obvious weaknesses in technical parts. Although the proposed method is relatively simple and may lack significant technical novelty, I believe the new insight this paper provides is valuable.

According to the requirement of the ICML 2025, the author should add a impact statement section.

**Questions For Authors:**

See above

**Relation To Broader Scientific Literature:**

N/A

**Theoretical Claims:**

Yes

---

> ### Author Rebuttal · Authors · 2025-03-29
>
> **Common response:**
>
> First of all, we would like to thank our reviewers for their time spent reviewing our paper along with the insightful comments they provided. Their reviews highlight that our method is “highly efficient and scalable” while underlining its innovative nature and our comprehensive experiments. Moreover, reviewer X44i points out the novelty of the theoretical framework we developed to ensure worst-case coverage bounds of vanilla CP. We appreciate all these comments.
>
> To address common questions shared among reviewer AUox and reviewer 3iVr, we add the following experiments which support our initial results:
> - **Shared model comparison:** we run a comparison of robust CP methods on an identical model, see answer to reviewer AUox.
> - **Lipschitz bound tightness estimation:** we empirically evaluate the Lipschitz constant of our network by running adversarial attacks, see answer to reviewer 3iVr.
>
> **Reviewer response:**
>
> We thank the reviewer for their positive feedback and strong endorsement of our work. We have incorporated an impact statement, as well as additional experiments (shared model comparison and empirical tightness estimation) to further highlight the strengths of our approach. We hope these enhancements further clarify and strengthen our contribution.

---

> > ### Comment · Reviewer_XbCa · 2025-04-05
> >
> > Thank you for your response and the new experimental results. I have decided to keep my score and am inclined to recommend acceptance.

---

### Official Review · Reviewer_AUox · 2025-03-06

**Overall Recommendation:** 2

**Summary:**

- This paper proposes a novel method, lip-rcp, for efficient robust conformal prediction (CP) by leveraging Lipschitz-bounded neural networks. The key contributions include:

- Theoretical analysis: Deriving worst-case coverage bounds for vanilla CP under l2 adversarial attacks, valid simultaneously for all perturbation budgets.

- Efficient robust CP: Introducing a method to compute robust prediction sets using globally Lipschitz-constrained networks, achieving state-of-the-art performance in terms of set size and computational efficiency.

- Scalability: Demonstrating applicability to large-scale datasets (e.g., ImageNet) with negligible computational overhead compared to vanilla CP.

- The experiments validate the method on CIFAR-10, CIFAR-100, TinyImageNet, and ImageNet, showing superior results over existing approaches like VRCP and CAS.


## update after rebuttal

After rebuttal, I still worry about the potential limitation in terms of its scope of applicability and the rationality of the theory. I maintain my original score, and am inclined to recommend a Weak Reject.

**Claims And Evidence:**

The claims are generally supported by theoretical proofs and empirical results but also some problem exist. For example:

- Claim 1: The authors call their method a CP method, but highly rely on training a specific classifier, which contradicts the model-independence of the CP method.

- Claim 2: Although author stated "We derive the first sound coverage bounds for vanilla CP that are valid simultaneously across all attack levels", it does not give an explicit quantification form, only gives a definition of the coverage bound.

- Claim 3: The worst-case coverage bounds (Theorem 3.3) are rigorously proven in Appendix B, assuming input space convexity and continuity of the non-conformity score. But the convex assumptions are unreasonable for image data in experiment.

- Claim 4: Although they provide a method to compute lower and upper bounds on CP, the discussion about the tightness of  bounds based the method is missing;

- Claim 5: The efficiency of lip-rcp is evidenced by Table 1 and Figure 2, showing O(1) complexity for non-conformity scores. However, the training overhead of Lipschitz networks (10–20% longer, per Appendix E) is not compared to baseline models.

- Claim 6: ImageNet scalability (Table 2) is demonstrated, but CAS is evaluated on only 500 samples versus lip-rcp’s 50,000, raising concerns about fairness.

**Essential References Not Discussed:**

Recent works CP vulnerabilities under adaptive attacks (Liu et al., 2024) are missing.

**Experimental Designs Or Analyses:**

Strengths:
- Comprehensive experiments across datasets and comparison to VRCP/CAS.

Weaknesses:

- The experiments comparison use different backbone model (ResNet50 in CAS vs. ResNeXt in lip-rcp). potentially biasing results.

- The ImageNet comparison uses unequal evaluation sizes (500 vs. 50k), potentially biasing results.

**Methods And Evaluation Criteria:**

Strengths:
- The use of Lipschitz networks to bound adversarial score variations is innovative and aligns well with the goal of robust CP.
The evaluation on standard benchmarks (CIFAR, ImageNet) is appropriate.

Weaknesses:
- The method is highly dependent on the specific Lipschitz network, but the estimation of the Lipschitz constant of common SOTA models is NP-hard, and the Lipschitz network is hard to obtain,  so the practical application of the method is limited
- The experiment focus on l2-bounded attacks limits practical relevance, as $\ell_\infty$ and $\ell_1$-attacks are more common in adversarial ML.

**Other Comments Or Suggestions:**

NO.

**Other Strengths And Weaknesses:**

No.

**Questions For Authors:**

- The CP method is a model-independent and data distribution-free technique.
As the authors state “the CP approach has the advantage of being applicable to any model,
even pre-trained on a different dataset” in introduction section. But the authors' proposed method relies on training
a specific 1-Lipschitz network which contradicts the model-independence of the CP approach.
How does it perform against other CP method based on the same backbone model, such as ResNet 50 e.t.?
How much influence does the structure of 1-Lipschitz network have on the effect?
For example, what is the result of CAS/RSCP method  combined with 1-Lipschitz network?

- Although the author mention the eq.19 is a estimate of Def 3.1. However, the Def 3.1 is defined in terms of upper (lower) infimum.
As the authors state "the informativeness of the bound is
directly linked to the tightness of the estimate" in Page 5.
An in-depth discussion for the tightness of eq.19 is missing.

-  Despite the author providing a definition of the Conservative/Restrictive Prediction Set in Def. 3.1,
further clarification is required regarding the practical application of this set of predictions.
It is necessary to elucidate the conditions under which the Conservative Prediction Set and the Restrictive Prediction Set are employed.
Additionally, the methods employed by the authors to calculate the prediction set in the experiment must be thoroughly explained.

- The experiments of work is focused on l2 attack, the l1 and l∞ results is missing, which is common in adversarial attacks.
The performance results of the method in these scenarios should be shown.

- Why is CAS evaluated on only 500 ImageNet samples but the lip-cp use 50000 samples in Table 2?
Would results hold with balanced sample sizes?

**Relation To Broader Scientific Literature:**

The work effectively bridges robust ML (Lipschitz networks) and uncertainty quantification (conformal prediction). It appropriately cites foundational CP works (Vovk et al., 2005) and recent robust CP methods  (Gendler et al., 2022; Jeary et al., 2024).

**Theoretical Claims:**

The proofs in Appendix B assume convex and closed input spaces (Assumption 3.2). But it is not valid for pixel-space images, which are not verified to be  convex and closed.

---

> ### Author Rebuttal · Authors · 2025-03-29
>
> **Common response:**
>
> First of all, we would like to thank our reviewers for their time spent reviewing our paper along with the insightful comments they provided. Their reviews highlight that our method is “highly efficient and scalable” while underlining its innovative nature and our comprehensive experiments. Moreover, reviewer X44i points out the novelty of the theoretical framework we developed to ensure worst-case coverage bounds of vanilla CP. We appreciate all these comments.
>
> To address common questions shared among reviewer AUox and reviewer 3iVr, we add the following experiments which support our initial results:
> - **Shared model comparison:** we run a comparison of robust CP methods on an identical model, see answer to reviewer AUox.
> - **Lipschitz bound tightness estimation:** we empirically evaluate the Lipschitz constant of our network by running adversarial attacks, see answer to reviewer 3iVr.
>
>
> **Reviewer response:**
>
> Foremost, we would like to thank Reviewer AUox for their comprehensive comments. We appreciate the comments about our methods' novelty and empirical validation.
>
> In the following paragraphs, we address their main concerns.
>
> **About comparing with different models (experimental weakness 1):**
>
> “The experiments comparison use different backbone model [...] potentially biasing results”
>
> To alleviate these concerns: we train a VGG-like 1-LipNet with 10.7M parameters from Boissin et al. 2025 on CIFAR-10. Then, we evaluate robust CP methods on this network under the conditions of Figure 3 .
>
> |Method|Coverage|Set size|Runtime(1 run)|
> |---|---|---|---|
> |CAS n_mc=1024 |94.6%|2.302|2615s|
> |CAS n_mc=10000|93.8%|2.057|5+ hours|
> |lip-rcp|**92.83%**|**1.889**|**10s**|
> |VRCP-I/C (CROWN)|OOM|OOM|N/A|
>
> CROWN does not scale to such a deep network due to its inner complexity.
>
> Therefore:
> - Using a deeper network further improves our results (10.7M parameters compared to 4.5M parameters for the ResNeXt).
> - LipNets allow for tighter conservative score estimations than CAS since its certificate is deterministic and does not require finite sample corrections.
> - This LipNet makes the CAS method perform better than with a ResNet50 due to its robustness (cf. Figure 3).
>
> These results will be added to the final version of the paper. We thank the reviewers for their comments which prompted this evaluation highlighting the performance and efficiency of our method.
>
> **About the ImageNet split sizes (experimental weakness 2):**
>
> To validate lip-rcp's performance without calibration set size differences we evaluate our method using only 500 calibration samples as done in the CAS article due to the method's high computational demands:
>
> Set sizes: 118.5  (111.0 originally)
> Coverage : 97.6% (97.4% originally)
>
> The performance gap between methods remains.
>
> **Theoretical questions (Claims 2 & 3):**
>
> While the manifold of images encountered in practice is typically nonconvex, Theorem 3.3 only requires that the input distribution and the score $x \mapsto s(x,y)$ be defined on a subset of a convex space such as $\mathcal{X}=[0,1]^{3 n_{\mathrm{pix}}}$, where $n_{\mathrm{pix}}$ is the number of pixels. Theorem 3.3 is then valid for any such distribution, even with finite support included in that space $\mathcal{X}$. We will add this fact within footnote 3 of Page 5.
>
> Also, our coverage bounds (15) are not closed-form, but (as mentioned after (15)) they can be quickly and tightly computed via a binary search. See also Langford and Schapire (2005, after Def 3.2) for closed-form yet looser bounds.
>
> **About model independence (Claim 1):**
>
> As correctly pointed out by the reviewer, we use additional information on the model to improve over black-box (model-free) approaches. We propose to insert the following clarification in Section 4:
>
> ``In practice, our method uses Lipschitz-by-design networks. This limits the model independence of our robust CP method, but it was key to obtaining efficient and competitive robust CP metrics for the first time. Interestingly, our methodology, which applies more generally to any network and score that are Lipschitz continuous, can also benefit from future research on Lipschitz constant estimation.''
>
> **About computational efficiency (Claim 5):**
>
> With several recent developments mentioned in the related works of our article, Lipschitz-by-design networks have become easier to train, and well performing, and libraries exist to train LipNets with minimal effort (cf geotorch, deel-lip, etc…).
> Training overheads are described in Appendix E, for an exhaustive study of the overheads of LipNets, we refer the reviewer to (Boissin et al. 2025, Table 3).
>
> For reference, our ResNeXt model of Figure 3 introduces a 9.6% runtime overhead on TinyImageNet compared to an identical unconstrained model.
>
> **Details:**
> The article by Liu et al. (2024) mentioned by the Reviewer is already cited in Page 1 L46.

---

> > ### Comment · Reviewer_AUox · 2025-04-04
> >
> > Thanks for the rebuttal. Unfortunately, the rebuttal has reinforced, rather than resolved, my concerns.
> > - Limited scope of practice: First, The CP method are known for model-independent and data distribution-free. But the authors' proposed method relies on training a specific 1-Lipschitz network which contradicts the model independence of the CP approach. Second, although author states in contribution that " We derive ... across all attack levels.", but the experiments of work is focused on l2 attack, the l1 and l∞ results is missing, which is common in adversarial attacks. Therefore, its scope of application is limited
> > - Limitations of the Theory: The author states that "Theorem 3.3 only requires that the input distribution and the score be defined on a subset of a convex space ", but the image is also not satisfied with the condition. Although it has a certain effect in practical performance, its theory can not explain the reason for its effect well.

---

> > > ### Author Response · Authors · 2025-04-04
> > >
> > > First of all, we would like to thank the reviewer for their reactivity.
> > >
> > > **About our scope of applicability:**
> > >
> > > As previously argued in our rebuttal, our methodology applies more generally to most SOTA networks that are Lipschitz continuous, not only *“on training a specific 1-Lipschitz network”*. Furthermore, our method can also benefit from future research on Lipschitz constant estimation.
> > >
> > > Perhaps more importantly, all competing approaches suffer from strong implicit requirements on the model. The high memory overhead of smoothing methods limits their applicability to small to medium models in practice at inference time. Also, verification methods often do not scale to deep networks.
> > >
> > > **About l_1 and l_inf robustness**
> > >
> > > Although we do not benchmark against l_1 or l_inf adversarial attacks (as stated in our limitations), it is clearly stated throughout the paper that our worst-case vanilla CP bounds are valid for any verifiable network (see Figure 4 left). This includes networks for l_1 and l_inf attacks. We will add relevant references.
> > >
> > > **About theoretical limitations:**
> > >
> > > We are quite confused about the concerns of the reviewer. Indeed, our theorem holds true for any convex space $\mathcal{X}$ that **contains** the image space and on which the score $x \mapsto s(x,y)$ is continuous. This property is straightforward in our case (with $\mathcal{X} = [0,1]^{3n_{pix}}$).

---

### Official Review · Reviewer_3iVr · 2025-03-12

**Overall Recommendation:** 2

**Summary:**

This paper addresses the limitations of robust conformal prediction (CP) under adversarial attacks. Traditional robust CP methods typically generate prediction sets that are either excessively large or computationally expensive for large-scale scenarios. To tackle these challenges, the authors introduce lip-rcp, which leverages Lipschitz-bounded neural networks to estimate robust prediction sets. By utilizing 1-Lipschitz constrained models, the proposed approach provides tighter and computationally efficient robust conformal prediction sets compared to existing methods.

**Claims And Evidence:**

Although the authors propose using Lipschitz-bounded neural networks to efficiently compute conservative and restrictive conformal scores, they acknowledge that accurately estimating the Lipschitz constant for deep neural networks remains computationally challenging and can result in overly conservative bounds if the estimates are loose. Thus, while their method claims efficiency and scalability, the precise tightness of these Lipschitz bounds and its corresponding impact over CP are not discussed in this paper.

**Essential References Not Discussed:**

This paper has discussed the essential works.

**Experimental Designs Or Analyses:**

1. Could you provide additional results on how accurate the estimation of the Lipschitz constant of the neural network?

2. How does lip-rcp perform in terms of robustness and accuracy trade-offs under different attack models?

**Methods And Evaluation Criteria:**

Equation (20) modifies the standard LAC conformity score by replacing softmax with sigmoid. Since softmax has no simple Lipschitz bound, sigmoid is used to maintain tractability. However, does this change affect the calibration of the non-conformity score? Could this lead to overly conservative or loose prediction sets depending on the logit scaling?

**Other Comments Or Suggestions:**

Some typos:

untractable $\rightarrow$ intractable

valid simultaneously for $\rightarrow$ valid simultaneously across

Tiny ImageNet $\rightarrow$ Tiny-ImageNet

**Other Strengths And Weaknesses:**

The key strength of this paper is its development of a highly efficient and scalable method, lip-rcp, that integrates Lipschitz-bounded neural networks into robust CP. Unlike previous robust CP methods, the proposed approach provides robust prediction sets with minimal computational cost. By leveraging networks designed with Lipschitz constraints, the authors achieve precise and certifiable bounds on prediction scores under adversarial perturbations.

**Questions For Authors:**

1. How tight are the Lipschitz bounds in practice? Have you quantified the potential gap between the estimated and actual Lipschitz constants?

2. Equation (20) modifies the standard LAC conformity score by replacing softmax with sigmoid. Since softmax has no simple Lipschitz bound, sigmoid is used to maintain tractability. However, does this change affect the calibration of the non-conformity score? Could this lead to overly conservative or loose prediction sets depending on the logit scaling?

3. Could you provide additional results on how accurate the estimation of the Lipschitz constant of the neural network?

4. Do the different choice of Lipschitz parametrized networks affect the coverage and efficiency of lip-rcp? Could you provide empirical results?

5. How does lip-rcp perform in terms of robustness and accuracy trade-offs under different attack models?

**Relation To Broader Scientific Literature:**

The key contributions of this paper build upon and extend several lines of research in robust conformal prediction (CP), adversarial robustness, and Lipschitz-bounded neural networks. Prior work on robust CP has primarily focused on randomized smoothing methods (Gendler et al., 2022; Yan et al., 2024) and formal verification-based approaches (Jeary et al., 2024), both of which provide robustness guarantees but suffer from either high computational costs or excessively large prediction sets. The paper improves upon these methods by introducing lip-rcp, inspired by work in certifiable adversarial robustness (Anil et al., 2019; Boissin et al., 2025).

**Theoretical Claims:**

1. Your method relies on 1-Lipschitz constrained networks. How does the setting of Lipschitz constant (e.g., different values of $L_n$) impact the robustness and efficiency of the prediction sets? Could you provide theoretical analysis?

2. Figure 5 is a good example of illustrating the proof. Could you provide more explanation of Figure 5 in the caption or appendix?

---

> ### Author Rebuttal · Authors · 2025-03-29
>
> **Common response:**
>
> First of all, we would like to thank our reviewers for their time spent reviewing our paper along with the insightful comments they provided. Their reviews highlight that our method is “highly efficient and scalable” while underlining its innovative nature and our comprehensive experiments. Moreover, reviewer X44i points out the novelty of the theoretical framework we developed to ensure worst-case coverage bounds of vanilla CP. We appreciate all these comments.
>
> To address common questions shared among reviewer AUox and reviewer 3iVr, we add the following experiments which support our initial results:
> - **Shared model comparison:** we run a comparison of robust CP methods on an identical model, see answer to reviewer AUox.
> - **Lipschitz bound tightness estimation:** we empirically evaluate the Lipschitz constant of our network by running adversarial attacks, see answer to reviewer 3iVr.
>
> **Reviewer response:**
>
> We would like to thank Reviewer 3iVr for their insightful comments. Importantly we appreciate their comments about our methods' efficiency, scalability and performance.
>
> Below we answer the reviewers main concerns.
>
> **About the tightness of Lipschitz bound estimations (Questions 1 & 3):**
>
> We run a direct tightness estimation of our Lipschitz upper bound with Lipschitz-by-design networks. We compute the maximum ratio between logit variations under attack and adversarial attack budgets on the CIFAR-10 test set.
>
> This quantity offers a lower bound to the actual Lipschitz-constant of our network.  Using PGD attacks of budget $\epsilon=0.05$ we get a Lipschitz constant lower bound of **0.917** when our by-design Lipschitz bound is **1**. This demonstrates that our bound is relatively tight in practice.
>
> Under these same attacks, the empirical coverage of our $\epsilon$-robust CP sets on the test split is 92.06% (94.7% under AutoAttack attacks) which approaches the desired robust coverage of 90% under $\alpha=0.1$.
>
> **About the choice of $L_n$ (Theoretical Claim 1):**
>
> To avoid any ambiguities, we address your question from two different angles.
> - Setting any $L_n \neq 1$ as the network constraint: this would have limited impact since constraining the Lipschitz constant of a classifier is not a limitation when using an appropriate optimization objective (Béthune et al. 2022). Moreover, $L_n = 1$  avoids gradient vanishing or explosion (Béthune et al. 2024, Thm 1).
> - Computing robust prediction sets using the l.h.s. of (19) and other values for $L_n$​: the approximation quality for the Lipschitz constant of the network is crucial, under-approximating it leads to uncertifiable results, while strongly over-estimating would lead to pathologically large prediction sets. Theory-wise, the impact of $L_n$​ can also be analyzed in a toy setting, with data points $(X_i,Y_i)$ drawn i.i.d. from a mixture of two Gaussians, and a model given by the Bayes rule.
>
> **About the LAC sigmoid score (Question 2):**
>
> Upon further inspection, the LAC sigmoid score yields a slight degradation of vanilla CP.
>
> CIFAR-10 / ResNet50 / $\alpha=0.1$:
> - LAC softmax coverage & set size: 90.04% / 1.088
> - LAC sigmoid coverage & set size: 90.28% / 1.144
>
> With similar tendencies for lower $\alpha$. We will add the following sentence to our paper:
>
> “This ablation study results in marginally bigger vanilla CP set sizes for LAC sigmoid scores compared to softmax ones with scaled temperature. Investigating Lipschitz conformal scores represents a promising direction for future research to further enhance robust CP performance.”
>
> **Regarding the tradeoff between accuracy and robustness (Theoretical claim 2 & Question 5):**
>
> Interestingly, the trade-off between accuracy and robustness for robust CP is not straightforward.
>
> Indeed, robust networks exhibit poorer accuracy which penalizes vanilla CP performance in small $\alpha$ regimes, consequently impacting robust CP performance. Similarly, accurate but brittle classifiers exhibit smaller vanilla CP set sizes yet the robust CP sets are large given the fine margins between conformal scores. To keep our method as simple, reliable and reproducible as possible we used standard hyperparameter values (which will be detailed in the Appendix) since our method exhibits SOTA behaviour without tuning.
>
> We would like to thank the reviewer for pointing out this promising aspect.
>
> **Details:**
>
> We will fix the typos that Reviewer 3iVr kindly pointed out. Furthermore, we propose the following caption for Figure 5:
>
> "Illustration of the proof. On the ball $\mathcal{B}\_{\epsilon + \delta}(x) $, the score $x \mapsto s(x,y)$ is minimized at some $\tilde{x}$. On the smaller ball $\mathcal{B}_{\epsilon}(x)$, the minimum can only be larger, but not larger than $s(x'',y)$, which is close to $s(\tilde{x},y)$ by continuity of $s(\cdot,y)$".

---

### Official Review · Reviewer_X44i · 2025-03-21

**Overall Recommendation:** 3

**Summary:**

This paper uses 1-Lipschitz networks to estimate robust conformal prediction (CP) sets, leading to the new lip-rcp method. The proposed method achieves SOTA results in the size of the robust CP sets and computational efficiency. In addition, the authors also study vanilla CP under attack, and derive new worst-case coverage bounds of vanilla CP sets.

**Claims And Evidence:**

The authors claim that their proposed lip-rcp method achieves SOTA in robust CP. It seems to me that this claim is supported by their study.

**Essential References Not Discussed:**

I do not have a particular paper in mind that this paper misses citing.

**Experimental Designs Or Analyses:**

The experiments are solid, and seem to support the claimed contribution.

**Methods And Evaluation Criteria:**

The empirical evaluation in this paper makes sense to me.

**Other Comments Or Suggestions:**

One thing which is not that clear to me is how to compare lip-rcp with other robust learning methods which are not based on CP in the first place.

**Other Strengths And Weaknesses:**

The paper is well written. The work is quite solid. The connection between Lipschitz networks and robust CP seems to be simple yet novel.

**Questions For Authors:**

Can the authors clarify what is the technical novelty of their theoretical contribution other than just merging Lipschitz networks with robust CP?

**Relation To Broader Scientific Literature:**

The topic studied in this paper seems quite relevant to the deep learning community. There are a lot of previous results on Lipschitz networks. This paper seems to be the first in leveraging such Lipschitz network results for studying robust CP. To me, this connection is novel and interesting, worth being known to the deep learning community.

**Theoretical Claims:**

The theorem look plausible to me, though I have not checked all the details carefully.

---

> ### Author Rebuttal · Authors · 2025-03-29
>
> **Common response:**
>
> First of all, we would like to thank our reviewers for their time spent reviewing our paper along with the insightful comments they provided. Their reviews highlight that our method is “highly efficient and scalable” while underlining its innovative nature and our comprehensive experiments. Moreover, reviewer X44i points out the novelty of the theoretical framework we developed to ensure worst-case coverage bounds of vanilla CP. We appreciate all these comments.
>
> To address common questions shared among reviewer AUox and reviewer 3iVr, we add the following experiments which support our initial results:
> - **Shared model comparison:** we run a comparison of robust CP methods on an identical model, see answer to reviewer AUox.
> - **Lipschitz bound tightness estimation:** we empirically evaluate the Lipschitz constant of our network by running adversarial attacks, see answer to reviewer 3iVr.
>
> **Reviewer response:**
>
> We would like to thank Reviewer X44i for the time dedicated to reviewing our paper and the suggestions they provided. Also, we appreciate the reviewer's comments regarding the soundness and novelty of our work.
>
> We develop answers to the reviewer's questions below:
>
> **About the connection with robust learning:**
>
> The field of Robust Conformal Prediction is quite specific, as it studies the robustness of *guaranteed prediction sets* under i.i.d conditions. It is not directly related to robust learning beyond reusing properties of a neural network, as the (split) Conformal procedure is post-hoc, and conducted on an already trained model.
>
> However, a comparison to any certifiably (or not) robust prediction set is feasible, based on other theories or heuristics. Yet, to our knowledge, robust prediction sets have only been studied in the Conformal Prediction setting.
>
> **About our theoretical contributions:**
>
> While one aspect of our work concerns developing a highly efficient method to compute certifiably robust prediction sets (lip-rcp) in the classic Robust CP setting of RSCP (Gendler et al., 2022). Our theoretical work of Section 3 introduces a novel complementary theoretical approach:
>
> In essence, we conduct a vanilla CP procedure (without robust score computations) and use an additional holdout data split to estimate maximum coverage variations for worst-case attacks. Those estimates are guaranteed for any budget simultaneously (uniformly). As opposed to Robust CP, this implies that under normal conditions, our method allows to retain the superior informativeness of vanilla CP sets while having an associated guarantee on how significantly their coverage under attack may evolve. As mentioned in the article, two previous works have proposed a weaker although incorrect guarantee, but with a similar intuition. Finally, the general formulation of our theoretical work allows us to extend our method to formal verification solvers (cf. Fig 4 - left).

---

### Decision · Program_Chairs · 2025-05-01

**Decision:**

Accept (poster)

**Comment:**

Two reviewers recommend acceptance and two reviewers recommend rejection.

Reviewer AUox that recommends a weak reject raises a few issues most of which seems to be resolved. Some of the remaining issues are not problematic in my opinion and do not warrant rejection. Specifically:
- Model-independence: While this is desirable it is not necessary, providing guarantees only for Lipschitz networks is interesting enough. However, I do agree that training "good" Lipschitz networks is hard in general, which somewhat limits the applicability. This must be clearly highlighted as a limitation by the authors in the final version.
- Convexity: As the authors explained in the rebuttal their result does not depend on the image space being convex or not, since assumption A1 holds. However, I recommend that the authors spend some time clarifying this matter in the paper (e.g. appendix) to avoid future confusion. Similarly, they should discuss when and why A2 holds for the models they consider.

In my opinion, the issues raised by Reviewer 3iVr who also recommends weak reject have been sufficiently addressed in the rebuttal, even though the reviewer did not change the score.

Another limitation that should be discussed and highlighted by the authors is the magnitude of the perturbation. Here $\epsilon$ ranges up to 0.05, while in the smoothing-based baselines (RSCP+, CAS) it is one order of magnitude higher (up to 0.5 and more).

Taken all together, I believe the paper has enough technical and experimental contributions to warrant acceptance.